# Dropping Symmetry for Fast Symmetric Nonnegative Matrix Factorization

**Zhihui Zhu**[*]
Mathematical Institute for Data Science
Johns Hopkins University
Baltimore, MD, USA
zzhu29@jhu.edu

**Xiao Li**[*]
Department of Electronic Engineering
The Chinese University of Hong Kong
Shatin, NT, Hong Kong
xli@ee.cuhk.edu.hk

**Kai Liu**
Department of Computer Science
Colorado School of Mines
Golden, CO, USA
kaliu@mines.edu

**Qiuwei Li**
Department of Electrical Engineering
Colorado School of Mines
Golden, CO, USA
qiuli@mines.edu

## Abstract

Symmetric nonnegative matrix factorization (NMF)—a special but important class of the general NMF—is demonstrated to be useful for data analysis and in particular for various clustering tasks. Unfortunately, designing fast algorithms for Symmetric NMF is not as easy as for the nonsymmetric counterpart, the later admitting the splitting property that allows efficient alternating-type algorithms. To overcome this issue, we transfer the symmetric NMF to a nonsymmetric one, then we can adopt the idea from the state-of-the-art algorithms for nonsymmetric NMF to design fast algorithms solving symmetric NMF. We rigorously establish that solving nonsymmetric reformulation returns a solution for symmetric NMF and then apply fast alternating based algorithms for the corresponding reformulated problem. Furthermore, we show these fast algorithms admit strong convergence guarantee in the sense that the generated sequence is convergent at least at a sublinear rate and it converges globally to a critical point of the symmetric NMF. We conduct experiments on both synthetic data and image clustering to support our result.

## 1 Introduction

General nonnegative matrix factorization (NMF) is referred to the following problem: Given a matrix $\boldsymbol{Y} \in \mathbb{R}^{n \times m}$ and a factorization rank $r$, solve

$$\min_{\boldsymbol{U} \in \mathbb{R}^{n \times r}, \boldsymbol{V} \in \mathbb{R}^{m \times r}} \frac{1}{2} \|\boldsymbol{Y} - \boldsymbol{U}\boldsymbol{V}^{\mathrm{T}}\|_F^2, \quad \text{s.t. } \boldsymbol{U} \geq \mathbf{0}, \boldsymbol{V} \geq \mathbf{0}, \tag{1}$$

where $\boldsymbol{U} \geq \mathbf{0}$ means each element in $\boldsymbol{U}$ is nonnegative. NMF has been successfully used in the applications of face feature extraction [1, 2], document clustering [3], source separation [4] and many others [5]. Because of the ubiquitous applications of NMF, many efficient algorithms have been proposed for solving (1). Well-known algorithms include MUA [6], projected gradientd descent [7], alternating nonnegative least squares (ANLS) [8], and hierarchical ALS (HALS) [9]. In particular, ANLS (which uses the block principal pivoting algorithm to very efficiently solve the nonnegative least squares) and HALS achive the state-of-the-art performance.

---

[*]Equal contribution

One special but important class of NMF, called symmetric NMF, requires the two factors $U$ and $V$ identical, i.e., it factorizes a PSD matrix $X \in \mathbb{R}^{n \times n}$ by solving

$$\min_{U \in \mathbb{R}^{n \times r}} \frac{1}{2} \|X - UU^{\mathrm{T}}\|_F^2, \quad \text{s.t. } U \geq 0. \tag{2}$$

As a contrast, (1) is referred to as nonsymmetric NMF. Symmetric NMF (2) has its own applications in data analysis, machine learning and signal processing [10, 11, 12]. In particular the symmetric NMF is equivalent to the classical $K$-means kernel clustering in [11]and it is inherently suitable for clustering nonlinearly separable data from a similarity matrix [10].

In the first glance, since (2) has only one variable, one may think it is easier to solve (2) than (1), or at least (2) can be solved by directly utilizing efficient algorithms developed for nonsymmetric NMF. However, the state-of-the-art alternating based algorithms (such as ANLS and HALS) for nonsymmetric NMF utilize the splitting property of (1) and thus can not be used for (2). On the other hand, first-order method such as projected gradient descent (PGD) for solving (2) suffers from very slow convergence. As a proof of concept, we show in Figure 1 the convergence of PGD for solving symmetric NMF and as a comparison, the convergence of gradient descent (GD) for solving a matrix factorization (MF) (i.e., (2) without the nonnegative constraint) which is proved to admit linear convergence [13, 14]. This phenomenon also appears in nonsymmetric NMF and is the main motivation to have many efficient algorithms such as ANLS and HALS.

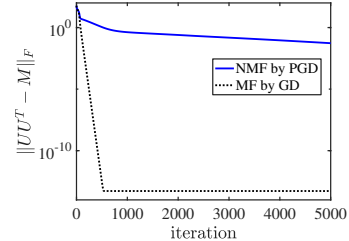

Figure 1: Convergence of MF by GD and symmetric NMF by PGD with the same initialization.

**Main Contributions**    This paper addresses the above issue by considering a simple framework that allows us to design alternating-type algorithms for solving the symmetric NMF, which are similar to alternating minimization algorithms (such as ANLS and HALS) developed for nonsymmetric NMF. The main contributions of this paper are summarized as follows.

- Motivated by the splitting property exploited in ANLS and HALS algorithms, we split the bilinear form of $U$ into two different factors and transfer the symmetric NMF into a nonsymmetric one:

$$\min_{U, V} f(U, V) = \frac{1}{2} \|X - UV^{\mathrm{T}}\|_F^2 + \frac{\lambda}{2} \|U - V\|_F^2, \quad \text{s.t. } U \geq 0, V \geq 0, \tag{3}$$

  where the regularizer $\|U - V\|_F^2$ is introduced to force the two factors identical and $\lambda > 0$ is a balancing factor. The first main contribution is to guarantee that any *critical point* of (3) that has bounded energy satisfies $U = V$ with a sufficiently large $\lambda$. We further show that any local-search algorithm with a decreasing property is guaranteed to solve (2) by targeting (3). To the best of our knowledge, this is the first work to *rigorously* establish that symmetric NMF can be efficiently solved by fast alternating-type algorithms.

- Our second contribution is to provide convergence analysis for our proposed alternating-based algorithms solving (3). By exploiting the specific structure in (3), we show that our proposed algorithms (without any proximal terms and any additional constraints on $U$ and $V$ except the nonnegative constraint) is convergent. Moreover, we establish the point-wise global sequence convergence and show that the proposed alternating-type algorithms achieve at least a global sublinear convergence rate. Our sequence convergence result provides theoretical guarantees for the practical utilization of alternating-based algorithms directly solving (3) without any proximal terms or additional constraint on the factors which are usually needed to guarantee the convergence.

**Related Work**    Due to slow convergence of PGD for solving symmetric NMF, different algorithms have been proposed to efficiently solve (2), either in a direct way or similar to (3) by splitting the two factors. Vandaele et al. [15] proposed an alternating algorithm that cyclically optimizes over each element in $U$ by solving a nonnegative constrained nonconvex univariate fourth order polynomial minimization. A quasi newton second order method was used in [10] to directly solve the symmetric NMF optimization problem (2). However, both the element-wise updating approach and the second order method are observed to be computationally expensive in large scale applications. We will illustrate this with experiments in Section 4.

The idea of solving symmetric NMF by targeting (3) also appears in [10]. However, despite an algorithm used for solving (3), no other formal guarantee (such as solving (3) returns a solution of (2)) was provided in [10]. Lu et al. [16] considered an alternative problem to (3) that also enjoys the splitting property and utilized alternating direction method of multipliers (ADMM) algorithm to tackle the corresponding problem with equality constraint (i.e., $U = V$). Unlike the sequence convergence guarantee of algorithms solving (3), the ADMM is only guaranteed to have a subsequence convergence in [16] with an additional proximal term[2] and constraint on the boundedness of columns of $U$, rendering the problem hard to solve.

A different line of research that may be also related to our work is the classical result on exact penalty methods [17, 18].[3] On one hand, our work shares similar spirit to the exact penalty methods as both approaches attempt to transfer constraints as penalties into the objective function. On the other hand, our approach differs from the exact penalty methods in the following three folds: 1) as stated in [18], most of the results on exact penalty methods require the feasible set compact, which is not satisfied in our problem (2); 2) the exact penalty methods replace all the constraints with certain penalty functions, while our reformulated problem (3) still has the nonnegative constriant; 3) in general the exact penalty methods use either nonsmooth penalty functions or differentiable exact penalty functions which are based on replacing the conventional multipliers with continuously differentiable multiplier functions, while the penalty in (3) is smooth and the parameter $\lambda$ is fixed and is independent of $U$ and $V$. We comment that our result builds on the specific structure of the objective function in (3) while the exact penalty methods focus on general nonlinear programming problems.

Finally, our work is also closely related to recent advances in convergence analysis for alternating minimizations. The sequence convergence result for general alternating minimization with an additional proximal term was provided in [19]. When specified to NMF, as pointed out in [20], with the aid of this additional proximal term (and also an additional constraint to bound the factors), the convergence of ANLS and HALS can be established from [19, 21]. With similar proximal term and constraint, the subsequence convergence of ADMM for symmetric NMF was obtained in [16]. Although the convergence of these algorithms are observed without the proximal term and constraint (which are also not used in practice), these are in general necessary to formally show the convergence of the algorithms. For alternating minimization methods solving (3), without any additional constraint, we show the factors are indeed bounded through the iterations, and without the proximal term, the algorithms admit sufficient decreasing property. These observations then guarantee the sequence convergence of the original algorithms that are used in practice. The convergence result for algorithms solving (3) is not only limited to alternating-type algorithms, though we only consider these as they achieve state-of-the-art performance.

## 2 Guarantee when Transfering Symmetric NMF to Nonsymmetric NMF

Compared with (2), in the first glance, (3) is slightly more complicated as it has one more variable. However, because of this new variable, $f(U, V)$ is now strongly convex with respect to either $U$ or $V$, though it is still nonconvex in terms of the joint variable $(U, V)$. Moreover, the two decision variables $U$ and $V$ in (3) are well splitted, like the case in nonsymmetric NMF. This observation suggests an interesting and useful fact that (3) can be solved by designing tailored alternating minimization type algorithms developed for tackling nonsymmetric NMF. On the other hand, a theoretical question raised in the regularized form (3) is that we are not guaranteed $U = V$ and hence solving (3) is not equivalent to solving (2). In this section, we provide formal guarantee to assure that solving (3) (to a critical point) indeed gives a critical point solution of (2). Note that (3) is nonconvex and many local search algorithms are only guaranteed to converge to a critical point rather than the global solution. Thus, our goal is to guarantee that any critical point that the algorithms may converge to admits the property that the two factors $U$ and $V$ are identical and further that $U$ is a solution ( critical point) of the original symmetric NMF (2).

Before stating out the formal result, as an intuitive example, we first consider a simple case where $f(u, v) = (1 - uv)^2/2 + \lambda(u - v)^2/2$. Its derivative is $\partial_u f(u, v) = (uv - 1)v + \lambda(u - v)$, $\partial_v f(u, v) = (uv - 1)u - \lambda(u - v)$. Thus, any critical point of $f$ satisfies $(uv - 1)v + \lambda(u - v) = 0$ and $(uv - 1)u - \lambda(u - v) = 0$, which further indicates that $(u - v)(2\lambda + 1 - uv) = 0$. Therefore,

for any critical point $(u, v)$ such that $|uv| < 2\lambda + 1$, it must satisfy $u = v$. Although (3) is more complicated than this example as it also has nonnegative constraint, the following result establishes similar guarantee for (3).[4]

**Theorem 1.** *Suppose $(\boldsymbol{U}^\star, \boldsymbol{V}^\star)$ be any critical point of (3) satisfying $\|\boldsymbol{U}^\star \boldsymbol{V}^{\star \mathrm{T}}\| < 2\lambda + \sigma_n(\boldsymbol{X})$, where $\sigma_n(\cdot)$ denotes the $n$-th largest singular value. Then $\boldsymbol{U}^\star = \boldsymbol{V}^\star$ and $\boldsymbol{U}^\star$ is a critical point of* (2).

Towards interpreting Theorem 1, we note that for any $\lambda > 0$, Theorem 1 ensures a certain region (whose size depends on $\lambda$) in which each critical point of (3) has identical factors and also returns a solution for the original symmetric NMF (2). This further suggests the opportunity of choosing an appropriate $\lambda$ such that the corresponding region (i.e., all $(\boldsymbol{U}, \boldsymbol{V})$ such that $\|\boldsymbol{U}\boldsymbol{V}^\mathrm{T}\| < 2\lambda + \sigma_n(\boldsymbol{X})$) contains all the possible points that the algorithms will converge to. Towards that end, next result indicates that for any local search algorithms, if it decreases the objective function, then the iterates are bounded.

**Lemma 1.** *For any local search algorithm solving (3) with initialization $\boldsymbol{V}_0 = \boldsymbol{U}_0, \boldsymbol{U}_0 \geq 0$, suppose it sequentially decreases the objective value. Then, for any $k \geq 0$, the iterate $(\boldsymbol{U}_k, \boldsymbol{V}_k)$ generated by this algorithm satisfies*

$$\|\boldsymbol{U}_k\|_F^2 + \|\boldsymbol{V}_k\|_F^2 \leq \left(\frac{1}{\lambda} + 2\sqrt{r}\right)\|\boldsymbol{X} - \boldsymbol{U}_0\boldsymbol{U}_0^\mathrm{T}\|_F^2 + 2\sqrt{r}\|\boldsymbol{X}\|_F := B_0,$$
$$\|\boldsymbol{U}_k\boldsymbol{V}_k^\mathrm{T}\|_F \leq \|\boldsymbol{X} - \boldsymbol{U}_0\boldsymbol{V}_0^\mathrm{T}\|_F + \|\boldsymbol{X}\|_F. \tag{4}$$

There are two interesting facts regarding the iterates can be interpreted from (4). The first equation of (4) implies that both $\boldsymbol{U}_k$ and $\boldsymbol{V}_k$ are bounded and the upper bound decays when the $\lambda$ increases. Specifically, as long as $\lambda$ is not too close to zero, then the RHS in (4) gives a meaningful bound which will be used for the convergence analysis of local search algorithms in next section. In terms of $\boldsymbol{U}_k\boldsymbol{V}_k^\mathrm{T}$, the second equation of (4) indicates that it is indeed upper bounded by a quantity that is independent of $\lambda$. This suggests a key result that if the iterative algorithm is convergent and the iterates $(\boldsymbol{U}_k, \boldsymbol{V}_k)$ converge to a critical point $(\boldsymbol{U}^\star, \boldsymbol{V}^\star)$, then $\boldsymbol{U}^\star \boldsymbol{V}^{\star \mathrm{T}}$ is also bounded, irrespectively the value of $\lambda$. This together with Theorem 1 ensures that many local search algorithms can be utilized to find a critical point of (2) by choosing a sufficiently large $\lambda$.

**Theorem 2.** *Choose $\lambda > \frac{1}{2}\left(\|\boldsymbol{X}\|_2 + \left\|\boldsymbol{X} - \boldsymbol{U}_0\boldsymbol{U}_0^\mathrm{T}\right\|_F - \sigma_n(\boldsymbol{X})\right)$ for (3), where $\|\boldsymbol{X}\|_2$ means largest singular value of $\boldsymbol{X}$. For any local search algorithm solving (3) with initialization $\boldsymbol{V}_0 = \boldsymbol{U}_0$, if it sequentially decreases the objective value, is convergent, and converges to a critical point $(\boldsymbol{U}^\star, \boldsymbol{V}^\star)$ of (3), then we have $\boldsymbol{U}^\star = \boldsymbol{V}^\star$ and that $\boldsymbol{U}^\star$ is also a critical point of* (2).

Theorem 2 indicates that instead of directly solving the symmetric NMF (2), one can turn to solve (3) with a sufficiently large regularization parameter $\lambda$. The latter is very similar to the nonsymmetric NMF (1) and obeys similar splitting property, which enables us to utilize efficient alternating-type algorithms. In the next section, we propose alternating based algorithms for tackling (3) with strong guarantees on the descend property and convergence issue.

## 3 Fast Algorithms for Symmetric NMF with Guaranteed Convergence

In the last section, we have shown that the symmetric NMF (2) can be transfered to problem (3), the latter admitting splitting property which enables us to design alternating-type algorithms to solve symmetric NMF. Specifically, we exploit the splitting property by adopting the main idea in ANLS and HALS for nonsymmetric NMF to design fast algorithms for (3). Moreover, note that the objective function $f$ in (3) is strongly convex with respect to $\boldsymbol{U}$ (or $\boldsymbol{V}$) with fixed $\boldsymbol{V}$ (or $\boldsymbol{U}$) because of the regularized term $\frac{\lambda}{2}\|\boldsymbol{U} - \boldsymbol{V}\|_F^2$. This together with Lemma 1 ensures that strong descend property and point-wise sequence convergence guarantee of the proposed alternating-type algorithms. With Theorem 2, we are finally guaranteed that the algorithms converge to a critical point of symmetric NMF (2).

### 3.1 ANLS for symmetric NMF (SymANLS)

**Algorithm 1** SymANLS

---

**Initialization:** $k = 1$ and $\boldsymbol{U}_0 = \boldsymbol{V}_0$.

1: **while** stop criterion not meet **do**
2: $\quad \boldsymbol{U}_k = \arg\min_{\boldsymbol{V} \geq 0} \frac{1}{2}\|\boldsymbol{X} - \boldsymbol{U}\boldsymbol{V}_{k-1}^{\mathrm{T}}\|_F^2 + \frac{\lambda}{2}\|\boldsymbol{U} - \boldsymbol{V}_{k-1}\|_F^2$;
3: $\quad \boldsymbol{V}_k = \arg\min_{\boldsymbol{U} \geq 0} \frac{1}{2}\|\boldsymbol{X} - \boldsymbol{U}_k\boldsymbol{V}^{\mathrm{T}}\|_F^2 + \frac{\lambda}{2}\|\boldsymbol{U}_k - \boldsymbol{V}\|_F^2$;
4: $\quad k = k + 1$.
5: **end while**

**Output:** factorization $(\boldsymbol{U}_k, \boldsymbol{V}_k)$.

---

ANLS is an alternating-type algorithm customized for nonsymmetric NMF (1) and its main idea is that at each time, keep one factor fixed, and update another one via solving a nonnegative constrained least squares. We use similar idea for solving (3) and refer the corresponding algorithm as SymANLS. Specifically, at the $k$-th iteration, SymANLS first updates $\boldsymbol{U}_k$ by

$$\boldsymbol{U}_k = \argmin_{\boldsymbol{U} \in \mathbb{R}^{n \times r}, \boldsymbol{U} \geq 0} \|\boldsymbol{X} - \boldsymbol{U}\boldsymbol{V}_{k-1}^{\mathrm{T}}\|_F^2 + \frac{\lambda}{2}\|\boldsymbol{U} - \boldsymbol{V}_{k-1}\|_F^2. \tag{5}$$

$\boldsymbol{V}_k$ is then updated in a similar way. We depict the whole procedure of SymANLS in Algorithm 1. With respect to solving the subproblem (5), we first note that there exists a unique minimizer (i.e., $\boldsymbol{U}_k$) for (5) as it involves a strongly objective function as well as a convex feasible region. However, we note that because of the nonnegative constraint, unlike least squares, in general there is no closed-from solution for (5) unless $r = 1$. Fortunately, there exist many feasible methods to solve the nonnegative constrained least squares, such as projected gradient descend, active set method and projected Newton's method. Among these methods, a block principal pivoting method is remarkably efficient for tackling the subproblem (5) (and also the one for updating $\boldsymbol{V}$) [8].

With the specific structure within (3) (i.e., its objective function is strongly convex and its feasible region is convex), we first show that SymANLS monotonically decreases the function value at each iteration, as required in Theorem 2.

**Lemma 2.** *Let $\{(\boldsymbol{U}_k, \boldsymbol{V}_k)\}$ be the iterates sequence generated by Algorithm 1. Then we have*

$$f(\boldsymbol{U}_k, \boldsymbol{V}_k) - f(\boldsymbol{U}_{k+1}, \boldsymbol{V}_{k+1}) \geq \frac{\lambda}{2}(\|\boldsymbol{U}_{k+1} - \boldsymbol{U}_k\|_F^2 + \|\boldsymbol{V}_{k+1} - \boldsymbol{V}_k\|_F^2).$$

We now give the following main convergence guarantee for Algorithm 1.

**Theorem 3** (Sequence convergence of Algorithm 1)**.** *Let $\{(\boldsymbol{U}_k, \boldsymbol{V}_k)\}$ be the sequence generated by Algorithm 1. Then*

$$\lim_{k \to \infty} (\boldsymbol{U}_k, \boldsymbol{V}_k) = (\boldsymbol{U}^\star, \boldsymbol{V}^\star),$$

*where $(\boldsymbol{U}^\star, \boldsymbol{V}^\star)$ is a critical point of (3). Furthermore the convergence rate is at least sublinear.*

Equipped with all the machinery developed above, the global sublinear sequence convergence of SymANLS to a critical solution of symmetric NMF (2) is formally guaranteed in the following result, which is a direct consequence of Theorem 2, Lemma 2 and Theorem 3.

**Corollary 1** (Convergence of Algorithm 1 to a critical point of (2))**.** *Suppose Algorithm 1 is initialized with $\boldsymbol{V}_0 = \boldsymbol{U}_0$. Choose*

$$\lambda > \frac{1}{2}\left(\|\boldsymbol{X}\|_2 + \left\|\boldsymbol{X} - \boldsymbol{U}_0\boldsymbol{U}_0^{\mathrm{T}}\right\|_F - \sigma_n(\boldsymbol{X})\right).$$

*Let $\{(\boldsymbol{U}_k, \boldsymbol{V}_k)\}$ be the sequence generated by Algorithm 1. Then $\{(\boldsymbol{U}_k, \boldsymbol{V}_k)\}$ is convergent and converges to $(\boldsymbol{U}^\star, \boldsymbol{V}^\star)$ with $\boldsymbol{U}^\star = \boldsymbol{V}^\star$ and $\boldsymbol{U}^\star$ a critical point of (2). Furthermore, the convergence rate is at least sublinear.*

**Remark.** We emphasis that the specific structure within (3) enables Corollary 1 get rid of the assumption on the boundedness of iterates $(\boldsymbol{U}_k, \boldsymbol{V}_k)$ and also the requirement of a proximal term, which is usually required for convergence analysis but not necessarily used in practice. As a contrast and also as pointed out in [20], to provide the convergence guarantee for standard ANLS solving nonsymmetric NMF (1), one needs to modify it by adding an additional proximal term as well as an additional constraint to make the factors bounded.

## 3.2 HALS for symmetric NMF (SymHALS)

As we stated before, due to the nonnegative constraint, there is no closed-from solution for (5), although one may utilize some efficient algorithms for solving (5). As a contrast, there does exist a closed-form solution when $r = 1$. HALS exploits this observation by splitting the pair of variables $(\boldsymbol{U}, \boldsymbol{V})$ into columns $(\boldsymbol{u}_1, \cdots, \boldsymbol{u}_r, \boldsymbol{v}_1, \cdots, \boldsymbol{v}_r)$ and then optimizing over *column by column*. We utilize similar idea for solving (3). Specifically, rewrite $\boldsymbol{U}\boldsymbol{V}^{\mathrm{T}} = \boldsymbol{u}_i \boldsymbol{v}_i^{\mathrm{T}} + \sum_{j \neq i} \boldsymbol{u}_j \boldsymbol{v}_j^{\mathrm{T}}$ and denote by

$$\boldsymbol{X}_i = \boldsymbol{X} - \sum_{j \neq i} \boldsymbol{u}_j \boldsymbol{v}_j^{\mathrm{T}}$$

the factorization residual $\boldsymbol{X} - \boldsymbol{U}\boldsymbol{V}^{\mathrm{T}}$ excluding $\boldsymbol{u}_i \boldsymbol{v}_i^{\mathrm{T}}$. Now if we minimize the objective function $f$ in (3) only with respect to $\boldsymbol{u}_i$, then it is equivalent to

$$\boldsymbol{u}_i^{\natural} = \arg \min_{\boldsymbol{u}_i \in \mathbb{R}^n} \frac{1}{2} \|\boldsymbol{X}_i - \boldsymbol{u}_i \boldsymbol{v}_i^{\mathrm{T}}\|_F^2 + \frac{\lambda}{2} \|\boldsymbol{u}_i - \boldsymbol{v}_i\|_2^2 = \max \left( \frac{(\boldsymbol{X}_i + \lambda \mathbf{I}) \boldsymbol{v}_i}{\|\boldsymbol{v}_i\|_2^2 + \lambda}, 0 \right).$$

Similar closed-form solution also holds when optimizing in terms of $\boldsymbol{v}_i$. With this observation, we utilize alternating-type minimization that at each time minimizes the objective function in (3) only with respect to one column in $\boldsymbol{U}$ or $\boldsymbol{V}$ and denote the corresponding algorithm as SymHALS. We depict SymHALS in Algorithm 2, where we use subscript $k$ to denote the $k$-th iteration. Note that to make the presentation easily understood, we directly use $\boldsymbol{X} - \sum_{j=1}^{i-1} \boldsymbol{u}_j^k (\boldsymbol{v}_j^k)^{\mathrm{T}} - \sum_{j=i+1}^{r} \boldsymbol{u}_j^{k-1} (\boldsymbol{v}_j^{k-1})^{\mathrm{T}}$ to update $\boldsymbol{X}_i^k$, which is not adopted in practice. Instead, letting $\boldsymbol{X}_1^k = \boldsymbol{X} - \boldsymbol{U}^{k-1} (\boldsymbol{V}^{k-1})^{\mathrm{T}}$, we can then update $\boldsymbol{X}_i^k$ with only the computation of $\boldsymbol{u}_i^k (\boldsymbol{v}_i^k)^{\mathrm{T}}$ by recursively utilizing the previous one. The detailed information about efficient implementation of SymHALS can be found in the supplementary material (see the corresponding Algorithm 3 in Section 5).

---

**Algorithm 2** SymHALS

**Initialization:** $\boldsymbol{U}_0, \boldsymbol{V}_0$, iteration $k = 1$.

1: **while** stop criterion not meet **do**
2:    **for** $i = 1 : r$ **do**
3:       $\boldsymbol{X}_i^k = \boldsymbol{X} - \sum_{j=1}^{i-1} \boldsymbol{u}_j^k (\boldsymbol{v}_j^k)^{\mathrm{T}} - \sum_{j=i+1}^{r} \boldsymbol{u}_j^{k-1} (\boldsymbol{v}_j^{k-1})^{\mathrm{T}}$;
4:       $\boldsymbol{u}_i^k = \arg \min_{\boldsymbol{u}_i \geq \mathbf{0}} \frac{1}{2} \|\boldsymbol{X}_i^k - \boldsymbol{u}_i (\boldsymbol{v}_i^{k-1})^{\mathrm{T}}\|_F^2 + \frac{\lambda}{2} \|\boldsymbol{u}_i - \boldsymbol{v}_i^{k-1}\|_2^2 = \max \left( \frac{(\boldsymbol{X}_i^k + \lambda \mathbf{I}) \boldsymbol{v}_i^{k-1}}{\|\boldsymbol{v}_i^{k-1}\|_2^2 + \lambda}, 0 \right)$;
5:       $\boldsymbol{v}_i^k = \arg \min_{\boldsymbol{v}_i \geq \mathbf{0}} \frac{1}{2} \|\boldsymbol{X}_i^k - \boldsymbol{u}_i^k \boldsymbol{v}_i^{\mathrm{T}}\|_F^2 + \frac{\lambda}{2} \|\boldsymbol{u}_i^k - \boldsymbol{v}_i\|_2^2 = \max \left( \frac{(\boldsymbol{X}_i^k + \lambda \mathbf{I}) \boldsymbol{u}_i^k}{\|\boldsymbol{u}_i^k\|_2^2 + \lambda}, 0 \right)$;
6:    **end for**
7:    $k = k + 1$.
8: **end while**

**Output:** factorization $(\boldsymbol{U}_k, \boldsymbol{V}_k)$.

---

The SymHALS enjoys similar descend property and convergence guarantee to algorithm SymANLS as both of them are alternating-based algorithms. Thus, we just give out the following results ensuring Algorithm 2 converge to a critical point of symmetric NMF (2).

**Corollary 2** (Convergence of Algorithm 2 to a critical point of (2)). *Suppose it is initialized with* $\boldsymbol{V}_0 = \boldsymbol{U}_0$. *Choose*

$$\lambda > \frac{1}{2} \left( \|\boldsymbol{X}\|_2 + \left\| \boldsymbol{X} - \boldsymbol{U}_0 \boldsymbol{U}_0^{\mathrm{T}} \right\|_F - \sigma_n(\boldsymbol{X}) \right).$$

*Let* $\{(\boldsymbol{U}_k, \boldsymbol{V}_k)\}$ *be the sequence generated by Algorithm 2. Then* $\{(\boldsymbol{U}_k, \boldsymbol{V}_k)\}$ *is convergent and converges to* $(\boldsymbol{U}^\star, \boldsymbol{V}^\star)$ *with* $\boldsymbol{U}^\star = \boldsymbol{V}^\star$ *and* $\boldsymbol{U}^\star$ *being a critical point of* (2). *Furthermore, the convergence rate is at least sublinear.*

**Remark.** Similar to Corollary 1 for SymANLS, Corollary 2 has no assumption on the boundedness of iterates $(\boldsymbol{U}_k, \boldsymbol{V}_k)$ and it establishes convergence guarantee for SymHALS without the aid from a proximal term. As a contrast, to establish the subsequence convergence for classical HALS solving nonsymmetric NMF [9, 22] (i.e., setting $\lambda = 0$ in SymHALS), one needs the assumption that every column of $(\boldsymbol{U}_k, \boldsymbol{V}_k)$ is not zero through all iterations. Though such assumption can be satisfied by using additional constraints, it actually solves a slightly different problem than the original

nonsymmetric NMF (1). On the other hand, SymHALS overcomes this issue and admits sequence convergence because of the additional regularizer in (3).

We end this section by noting that both Theorem 2 and the convergence guarantee in Corollary 1 and Corollary 2 can be extended to the case with any additional convex constraint and/or regularizer on $U$. For example, for promoting sparsity, one can use $\ell_1$ constraint or regularizer which can also be efficiently incorporated into SymHALS or SymGCD. A formal guarantee for this extension is the subject of ongoing work.

## 4 Experiments on Synthetic Data and Real Image Data

In this section, we conduct experiments on both synthetic data and real data to illustrate the performance of our proposed algorithms and compare it to other state-of-the-art ones, in terms of both convergence property and image clustering performance.

For comparison convenience, we define

$$E^k = \frac{\|X - U^k(U^k)^{\mathrm{T}}\|_F^2}{\|X\|_F^2}$$

as the normalized fitting error at $k$-th iteration.

Besides SymANLS and SymHALS, we also apply the greedy coordinate descent (GCD) algorithm in [23] (which is designed for tackling nonsymmetric NMF) to solve the reformulated problem (3) and denote the corrosponding algorithm as **SymGCD**. SymGCD is expected to have similar sequence convergence guarantee as SymANLS and SymHALS. We list the algorithms to compare: 1) **ADMM** in [16], where there is a regularization item in their augmented Lagrangian and we tune a good one for comparison; 2) **SymNewton** [10] which is a Newton-like algorithm by with a the Hessian matrix in Newton's method for computation efficiency; and 3) **PGD** in [7]. The algorithm in [15] is inefficient for large scale data, since they apply an alternating minimization over each coordinate which entails many loops for large scale $U$.

### 4.1 Convergence verification

We randomly generate a matrix $U \in \mathbb{R}^{50 \times 5}(n = 50, r = 5)$ with each entry independently following a standard Gaussian distribution. To enforce nonnegativity, we then take absolute value on each entry of $U$ to get $U^\star$. Data matrix $X$ is constructed as $U^\star(U^\star)^{\mathrm{T}}$ which is nonnegative and PSD. We initialize all the algorithms with same $U^0$ and $V^0$, whose entries are *i.i.d.* uniformly distributed between 0 to 1.

To study the effect of the parameter $\lambda$ in (3), we show the value $\|U_k - V_k\|_F^2$ versus iteration for different choices of $\lambda$ by SymHALS in Figure 2. While for this experimental setting the lower bound of $\lambda$ provided in Theorem 2 is 39.9, we observe that $\|U_k - V_k\|_F^2$ still converges to 0 with much smaller $\lambda$. This suggests that the *sufficient condition* on the choice of $\lambda$ in Theorem 2 is stronger than necessary, leaving room for future improvements. Particularly, we suspect that SymHALS converges to a critical point $(U^\star, V^\star)$ with $U^\star = V^\star$ (i.e. a critical point of symmetric NMF) for any $\lambda > 0$; we leave this line of theoretical justification as our future work. On the other hand, we note that although SymHALS finds a critical point of symmetric NMF for most of the $\lambda$, the convergence speed varies for different $\lambda$. For example, we observe that either a very large or small $\lambda$ yields a slow convergence speed. In the sequel, we tune the best parameter $\lambda$ for each experiment.

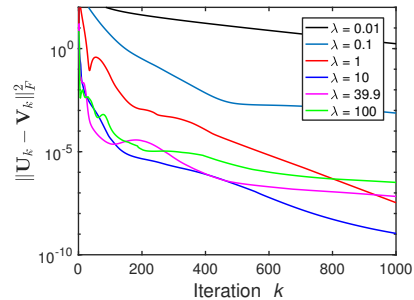

Figure 2: SymHALS with different $\lambda$. Here $n = 50, r = 5$.

We also test on real world dataset CBCL [5], where there are 2429 face image data with dimension $19 \times 19$. We construct the similarity matrix $\boldsymbol{X}$ following [10, section 7.1, step 1 to step 3]. The convergence results on synthetic data and real world data are shown in Figure 3 (a1)-(a2) and Figure 3 (b1)-(b2), respectively. We observe that the SymANLS, SymHALS, and SymGCD 1) converge faster; 2) empirically have a linear convergence rate in terms of $E^k$.

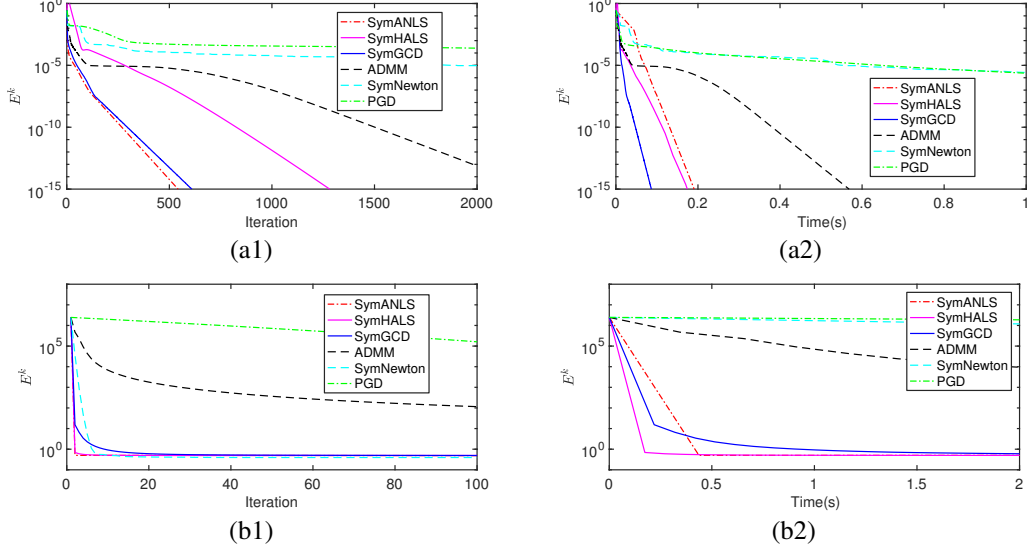

Figure 3: Synthetic data where $n = 50, r = 5$: (a1)-(a2) fitting error versus iteration and running time. Real image dataset CBCL where $n = 2429, r = 49$: (b1)-(b2) fitting error versus iteration and running time.

## 4.2 Image clustering

Symmetric NMF can be used for graph clustering [10, 11] where each element $\boldsymbol{X}_{ij}$ denotes the similarity between data $i$ and $j$. In this subsection, we apply different symmetric NMF algorithms for graph clustering on image datasets and compare the clustering accuracy [24].

We put all images to be clustered in a data matrix $\boldsymbol{M}$, where each row is a vectorized image. We construct similarity matrix following the procedures in [10, section 7.1, step 1 to step 3], and utilize self-tuning method to construct the similarity matrix $\boldsymbol{X}$. Upon deriving $\widetilde{\boldsymbol{U}}$ from symmetric NMF $\boldsymbol{X} \approx \widetilde{\boldsymbol{U}} \widetilde{\boldsymbol{U}}^{\mathrm{T}}$, the label of the $i$-th image can be obtained by:

$$l(\boldsymbol{M}_i) = \arg \max_{j} \widetilde{\boldsymbol{U}}_{(ij)}. \tag{6}$$

We conduct the experiments on four image datasets:

**ORL**: 400 facial images from 40 different persons with each one has 10 images from different angles and emotions [6].

**COIL-20**: 1440 images from 20 objects [7].

**TDT2**: 10,212 news articles from 30 categories [8]. We extract the first 3147 data for experiments (containing only 2 categories).

**MNIST**: classical handwritten digits dataset [9], where 60,000 are for training (denoted as $\mathrm{MNIST}_{train}$), and 10,000 for testing (denoted as $\mathrm{MNIST}_{test}$). we test on the first 3147 data from $\mathrm{MNIST}_{train}$ (contains 10 digits) and 3147 from $\mathrm{MNIST}_{test}$ (contains only 3 digits) .

In Figure 4 (a1) and Figure 4(a2), we display the clustering accuracy on dataset **ORL** with respect to iterations and time (only show first 10 seconds), respectively. Similar results for dataset **COIL-20** are plotted in Figure 4 (b1)-(b2). We observe that in terms of iteration number, SymNewton has comparable performance to the three alternating methods for (3) (i.e., SymANLS, SymHALS, and SymGCD), but the latter outperform the former in terms of running time. Such superiority becomes more apparent when the size of the dataset increases. We show the comparison on larger truncated datasets MNIST$_{train}$ and MNIST$_{test}$ in supplementary materials. We note that the performance of ADMM will increase as iterations goes and after almost 3500 iterations on ORL dataset it reaches a comparable result to other algorithms. Moreover, it requires more iterations for larger dataset. This observation makes ADMM not practical for image clustering. We run ADMM 5000 iterations on ORL dataset, and display it in the supplementary material. These results as well as the experimental results shown in the last subsection demonstrate $(i)$ the power of transfering the symmetric NMF (2) to a nonsymmetric one (3); and $(ii)$ the efficieny of alternating-type algorithms for sovling (3) by exploiting the splitting property within the optimization variables in (3).

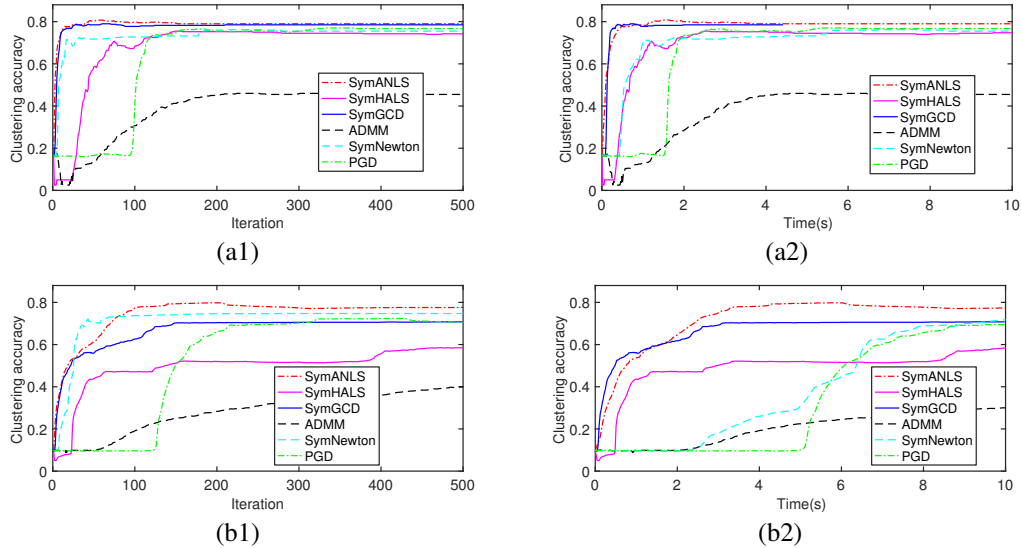

Figure 4: Real dataset: (a1) and (a2) Image clustering quality on ORL dataset, $n = 400, r = 40$; (b1) and (b2) Image clustering quality on COIL-20 dataset, $n = 1440, r = 20$.

Table 1 shows the clustering accuracies of different algorithms on different datasets, where we run enough iterations for ADMM so that it obtains its best result. We observe from Table 1 that SymANLS, SymHALS, and SymGCD perform better than or have comparable performance to others for most of the cases.

Table 1: Summary of image clustering accuracy of different algorithms on five image datasets

|  | ORL | COIL-20 | MNIST$_{train}$ | TDT2 | MNIST$_{test}$ |
|---|---|---|---|---|---|
| SymANLS | **0.8075** | **0.7979** | 0.6477 | 0.9800 | 0.8589 |
| SymHALS | 0.7550 | 0.5854 | **0.6657** | **0.9806** | 0.8608 |
| SymGCD | 0.7900 | 0.7076 | 0.6293 | 0.9803 | **0.9882** |
| ADMM | 0.7650 | 0.6903 | 0.5803 | 0.9800 | 0.8713 |
| SymNewton | 0.7625 | 0.7472 | 0.5990 | 0.9793 | 0.8589 |
| PGD | 0.7700 | 0.7243 | 0.6475 | 0.9800 | 0.8710 |

## Acknowledgment

The authors thank Dr. Songtao Lu for sharing the code used in [16] and the three anonymous reviewers as well as the area chair for their constructive comments.

## Footnotes

[2]In $k$-th iteration, a proximal term $\|U - U_{k-1}\|_F^2$ is added to the objective function when updating $U$.

[3]We greatly acknolwedge the anonymous area chair for pointing out this related work.

[4]All the proofs can be found at supplementary material.

[5]http://cbcl.mit.edu/software-datasets/FaceData2.html

[6]http://www.cl.cam.ac.uk/research/dtg/attarchive/facedatabase.html

[7]http://www.cs.columbia.edu/CAVE/software/softlib/coil-20.php

[8]https://www.ldc.upenn.edu/collaborations/past-projects

[9]http://yann.lecun.com/exdb/mnist/

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
