[Supplementary Material · supplementary_SNMF_nips2018.pdf]

# Supplemental Materials for: "Dropping Symmetry for Fast Symmetric Nonnegative Matrix Factorization"

## 1 Problem Statement and Transferring Symmetric NMF to Nonsymmetric NMF

**Problem Statement.** General nonnegative matrix factorization (NMF) is referred to the following problem: Given a matrix $\boldsymbol{Y} \in \mathbb{R}^{n \times m}$ and a factorization rank $r$, solve

$$\min_{\boldsymbol{U} \in \mathbb{R}^{n \times r}, \boldsymbol{V} \in \mathbb{R}^{m \times r}} \frac{1}{2}\|\boldsymbol{Y} - \boldsymbol{U}\boldsymbol{V}^{\mathrm{T}}\|_F^2, \quad \text{subject to } \boldsymbol{U} \geq \boldsymbol{0}, \boldsymbol{V} \geq \boldsymbol{0}, \tag{1}$$

where $\boldsymbol{U} \geq \boldsymbol{0}$ means each element in $\boldsymbol{U}$ is nonnegative.

When the two factors $\boldsymbol{U}$ and $\boldsymbol{V}$ are required identical, (2) becomes the following symmetric NMF: given a PSD matrix $\boldsymbol{X} \in \mathbb{R}^{n \times n}$

$$\min_{\boldsymbol{U}} \frac{1}{2}\|\boldsymbol{X} - \boldsymbol{U}\boldsymbol{U}^{\mathrm{T}}\|_F^2, \quad \text{subject to } \boldsymbol{U} \geq \boldsymbol{0}. \tag{2}$$

In the following, we call (1) nonsymmetric NMF.

**Problem Reformulation: Dropping Symmetry.** We transfer the symmetric NMF problem in (2) to

$$\min_{\boldsymbol{U}, \boldsymbol{V}} f(\boldsymbol{U}, \boldsymbol{V}) = \frac{1}{2}\|\boldsymbol{X} - \boldsymbol{U}\boldsymbol{V}^{\mathrm{T}}\|_F^2 + \frac{\lambda}{2}\|\boldsymbol{U} - \boldsymbol{V}\|_F^2, \quad \text{subject to } \boldsymbol{U} \geq \boldsymbol{0}, \boldsymbol{V} \geq \boldsymbol{0}. \tag{3}$$

To theoretically establish this transformation, we restate Theorem 1, Lemma 1, and Theorem 2 in the original paper and give their proofs.

**Theorem 1 (Restatement of Theorem 1 in the original paper).** *Suppose $(\boldsymbol{U}^\star, \boldsymbol{V}^\star)$ be any critical point of* (3) *satisfying $\|\boldsymbol{U}^\star \boldsymbol{V}^{\star\mathrm{T}}\| < 2\lambda + \sigma_n(\boldsymbol{X})$, where $\sigma_n(\cdot)$ denotes the n-th largest singular value. Then $\boldsymbol{U}^\star = \boldsymbol{V}^\star$ and $\boldsymbol{U}^\star$ is a critical point of* (2).

*Proof of Theorem 1.* We first preset the following useful result, which generalizes the classical result for two PSD matrices.

**Lemma 1.** *For any symmetric $\boldsymbol{A} \in \mathbb{R}^{n \times n}$ and PSD matrix $\boldsymbol{B} \in \mathbb{R}^{n \times n}$, we have*

$$\sigma_n(\boldsymbol{A})\operatorname{trace}(\boldsymbol{B}) \leq \operatorname{trace}(\boldsymbol{A}\boldsymbol{B}) \leq \sigma_1(\boldsymbol{A})\operatorname{trace}(\boldsymbol{B}),$$

*where $\sigma_i(\boldsymbol{A})$ is the i-th largest eigenvalue of $\boldsymbol{A}$.*

*Proof of Lemma 1.* Let $\boldsymbol{A} = \boldsymbol{\Phi}_1 \boldsymbol{\Lambda}_1 \boldsymbol{\Phi}_1^{\mathrm{T}}$ and $\boldsymbol{B} = \boldsymbol{\Phi}_2 \boldsymbol{\Lambda}_2 \boldsymbol{\Phi}_2^{\mathrm{T}}$ be the eigendecompositions of $\boldsymbol{A}$ and $\boldsymbol{B}$, respectively. Here $\boldsymbol{\Lambda}_1$ $(\boldsymbol{\Lambda}_2)$ is a diagonal matrix with the eigenvalues of $\boldsymbol{A}$ $(\boldsymbol{B})$ along its diagonal. We first rewrite $\operatorname{trace}(\boldsymbol{A}\boldsymbol{B})$ as

$$\operatorname{trace}(\boldsymbol{A}\boldsymbol{B}) = \operatorname{trace}\left(\boldsymbol{\Lambda}_1 \boldsymbol{\Phi}_1^{\mathrm{T}} \boldsymbol{\Phi}_2 \boldsymbol{\Lambda}_2 \boldsymbol{\Phi}_2^{\mathrm{T}} \boldsymbol{\Phi}_1\right).$$

Noting that $\boldsymbol{\Lambda}_1$ is a diagonal matrix and $\boldsymbol{\Phi}_1^{\mathrm{T}} \boldsymbol{\Phi}_2 \boldsymbol{\Lambda}_2 \boldsymbol{\Phi}_2^{\mathrm{T}} \boldsymbol{\Phi}_1 \succeq \boldsymbol{0}$ since $\boldsymbol{\Lambda}_2 \succeq \boldsymbol{0}$, we have

$$\operatorname{trace}\left(\boldsymbol{\Lambda}_1 \boldsymbol{\Phi}_1^{\mathrm{T}} \boldsymbol{\Phi}_2 \boldsymbol{\Lambda}_2 \boldsymbol{\Phi}_2^{\mathrm{T}} \boldsymbol{\Phi}_1\right) \leq \max_i \boldsymbol{\Lambda}_1[i, i] \cdot \operatorname{trace}\left(\boldsymbol{\Phi}_1^{\mathrm{T}} \boldsymbol{\Phi}_2 \boldsymbol{\Lambda}_2 \boldsymbol{\Phi}_2^{\mathrm{T}} \boldsymbol{\Phi}_1\right) = \sigma_1(\boldsymbol{A})\operatorname{trace}(\boldsymbol{B}).$$

The other direction follows similarly. $\qquad \square$

We now turn to proving Theorem 1. The subdifferential of $f$ is given as follows

$$\partial_{\boldsymbol{U}} f(\boldsymbol{U}, \boldsymbol{V}) = (\boldsymbol{U}\boldsymbol{V}^{\mathrm{T}} - \boldsymbol{X})\boldsymbol{V} + \lambda(\boldsymbol{U} - \boldsymbol{V}) + \partial\delta_+(\boldsymbol{U}), \tag{4}$$

$$\partial_{\boldsymbol{V}} f(\boldsymbol{U}, \boldsymbol{V}) = (\boldsymbol{U}\boldsymbol{V}^{\mathrm{T}} - \boldsymbol{X})^{\mathrm{T}}\boldsymbol{U} - \lambda(\boldsymbol{U} - \boldsymbol{V}) + \partial\delta_+(\boldsymbol{V}), \tag{5}$$

where $\partial\delta_+(\boldsymbol{U}) = \{\boldsymbol{G} \in \mathbb{R}^{n \times r} : \boldsymbol{G} \circ \boldsymbol{U} = \boldsymbol{0}, \boldsymbol{G} \leq \boldsymbol{0}\}$ when $\boldsymbol{U} \geq \boldsymbol{0}$ and otherwise $\partial\delta_+(\boldsymbol{U}) = \emptyset$. Since $(\boldsymbol{U}^\star, \boldsymbol{V}^\star)$ is a critical point of (3), it satisfies

$$(\boldsymbol{U}^\star\boldsymbol{V}^{\star\mathrm{T}} - \boldsymbol{X})\boldsymbol{V}^\star + \lambda(\boldsymbol{U}^\star - \boldsymbol{V}^\star) + \boldsymbol{G} = \boldsymbol{0}, \tag{6}$$

$$(\boldsymbol{U}^\star\boldsymbol{V}^{\star\mathrm{T}} - \boldsymbol{X})^{\mathrm{T}}\boldsymbol{U}^\star - \lambda(\boldsymbol{U}^\star - \boldsymbol{V}^\star) + \boldsymbol{H} = \boldsymbol{0}, \tag{7}$$

where $\boldsymbol{G} \in \partial\delta_+(\boldsymbol{U}^\star)$ and $\boldsymbol{H} \in \partial\delta_+(\boldsymbol{V}^\star)$. Subtracting (7) from (6), we have

$$(2\lambda\mathbf{I} + \boldsymbol{X})(\boldsymbol{U}^\star - \boldsymbol{V}^\star) = \boldsymbol{V}^\star\boldsymbol{U}^{\star\mathrm{T}}\boldsymbol{U}^\star - \boldsymbol{U}^\star\boldsymbol{V}^{\star\mathrm{T}}\boldsymbol{V}^\star - \boldsymbol{G} + \boldsymbol{H}. \tag{8}$$

where we utilize the fact that $\boldsymbol{X}$ is symmetric, i.e., $\boldsymbol{X} = \boldsymbol{X}^{\mathrm{T}}$. Taking the inner product of $\boldsymbol{U}^\star - \boldsymbol{V}^\star$ with both sides of the above equation gives

$$\langle(\lambda\mathbf{I} + \boldsymbol{X}), (\boldsymbol{U}^\star - \boldsymbol{V}^\star)(\boldsymbol{U}^\star - \boldsymbol{V}^\star)^{\mathrm{T}}\rangle = \langle\boldsymbol{V}^\star\boldsymbol{U}^{\star\mathrm{T}}\boldsymbol{U}^\star - \boldsymbol{U}^\star\boldsymbol{V}^{\star\mathrm{T}}\boldsymbol{V}^\star - \boldsymbol{G} + \boldsymbol{H}, \boldsymbol{U}^\star - \boldsymbol{V}^\star\rangle. \tag{9}$$

In what follows, by choosing sufficiently large $\lambda$, we show that $(\boldsymbol{U}^\star, \boldsymbol{V}^\star)$ satisfying (9) must satisfy $\boldsymbol{U}^\star = \boldsymbol{V}^\star$. To that end, we first provide the lower bound and the upper bound for the LHS and RHS of (9), respectively. Specifically,

$$\langle((2\lambda\mathbf{I} + \boldsymbol{X}), (\boldsymbol{U}^\star - \boldsymbol{V}^\star)(\boldsymbol{U}^\star - \boldsymbol{V}^\star)^{\mathrm{T}}\rangle \geq \sigma_n((2\lambda\mathbf{I} + \boldsymbol{X})\|\boldsymbol{U}^\star - \boldsymbol{V}^\star\|_F^2 = ((2\lambda + \sigma_n(\boldsymbol{X}))\|\boldsymbol{U}^\star - \boldsymbol{V}^\star\|_F^2, \tag{10}$$

where the inequality follows from Lemma 1. On the other hand,

$$
\begin{aligned}
&\langle\boldsymbol{V}^\star\boldsymbol{U}^{\star\mathrm{T}}\boldsymbol{U}^\star - \boldsymbol{U}^\star\boldsymbol{V}^{\star\mathrm{T}}\boldsymbol{V}^\star - \boldsymbol{G} + \boldsymbol{H}, \boldsymbol{U}^\star - \boldsymbol{V}^\star\rangle \\
&\leq \langle\boldsymbol{V}^\star\boldsymbol{U}^{\star\mathrm{T}}\boldsymbol{U}^\star - \boldsymbol{U}^\star\boldsymbol{V}^{\star\mathrm{T}}\boldsymbol{V}^\star, \boldsymbol{U}^\star - \boldsymbol{V}^\star\rangle \\
&= \left\langle \frac{\boldsymbol{V}^\star\boldsymbol{U}^{\star\mathrm{T}} + \boldsymbol{U}^\star\boldsymbol{V}^{\star\mathrm{T}}}{2}, (\boldsymbol{U}^\star - \boldsymbol{V}^\star)(\boldsymbol{U}^\star - \boldsymbol{V}^\star)^{\mathrm{T}} \right\rangle - \frac{1}{2}\left\|\boldsymbol{U}^\star\boldsymbol{V}^{\star\mathrm{T}} - \boldsymbol{V}^\star\boldsymbol{U}^{\star\mathrm{T}}\right\|_F^2 \\
&\leq \left\langle \frac{\boldsymbol{V}^\star\boldsymbol{U}^{\star\mathrm{T}} + \boldsymbol{U}^\star\boldsymbol{V}^{\star\mathrm{T}}}{2}, (\boldsymbol{U}^\star - \boldsymbol{V}^\star)(\boldsymbol{U}^\star - \boldsymbol{V}^\star)^{\mathrm{T}} \right\rangle \\
&\leq \sigma_1\left(\frac{\boldsymbol{V}^\star\boldsymbol{U}^{\star\mathrm{T}} + \boldsymbol{U}^\star\boldsymbol{V}^{\star\mathrm{T}}}{2}\right)\|\boldsymbol{U}^\star - \boldsymbol{V}^\star\|_F^2
\end{aligned} \tag{11}
$$

where the last inequality utilizes Lemma 1 and the first inequality follows because $\boldsymbol{V}^\star, \boldsymbol{U}^\star \geq \boldsymbol{0}$ indicating that

$$-\langle\boldsymbol{G}, \boldsymbol{U}^\star - \boldsymbol{V}^\star\rangle \leq 0, \quad \langle\boldsymbol{H}, \boldsymbol{U}^\star - \boldsymbol{V}^\star\rangle \leq 0$$

Now plugging (10) and (11) back into (9) and utilizing the assumption that $\|\boldsymbol{U}^\star\boldsymbol{V}^{\star\mathrm{T}}\|_F \leq \alpha$, we have

$$((2\lambda + \sigma_n(\boldsymbol{X}))\|\boldsymbol{U}^\star - \boldsymbol{V}^\star\|_F^2 \leq \sigma_1\left(\frac{\boldsymbol{V}^\star\boldsymbol{U}^{\star\mathrm{T}} + \boldsymbol{U}^\star\boldsymbol{V}^{\star\mathrm{T}}}{2}\right)\|\boldsymbol{U}^\star - \boldsymbol{V}^\star\|_F^2 \leq \alpha\|\boldsymbol{U}^\star - \boldsymbol{V}^\star\|_F^2,$$

which implies that if we choose $2\lambda > \alpha - \sigma_n(\boldsymbol{X})$, then $\boldsymbol{U}^\star = \boldsymbol{V}^\star$ must hold. Plugging it into (4) gives

$$\boldsymbol{0} \in (\boldsymbol{U}^\star(\boldsymbol{U}^\star)^{\mathrm{T}} - \boldsymbol{X})\boldsymbol{U}^\star + \partial\delta_+(\boldsymbol{U}^\star).$$

which implies $\boldsymbol{U}^\star$ is a critical point of (2). $\qquad\square$

**Lemma 2** (**Restatement of Lemma 1 in the original paper**). *For any local search algorithm solving* (3) *with initialization* $\boldsymbol{V}_0 = \boldsymbol{U}_0$, *suppose it sequentially decreases the objective value. Then, for any* $k \geq 0$, *the iterate* $(\boldsymbol{U}_k, \boldsymbol{V}_k)$ *generated by this algorithm satisfies*

$$\|\boldsymbol{U}_k\|_F^2 + \|\boldsymbol{V}_k\|_F^2 \leq \left(\frac{1}{\lambda} + 2\sqrt{r}\right) \|\boldsymbol{X} - \boldsymbol{U}_0\boldsymbol{U}_0^T\|_F^2 + 2\sqrt{r}\|\boldsymbol{X}\|_F := B_0,$$

$$\|\boldsymbol{U}_k\boldsymbol{V}_k^T\|_F \leq \|\boldsymbol{X} - \boldsymbol{U}_0\boldsymbol{V}_0^T\|_F + \|\boldsymbol{X}\|_F. \tag{12}$$

*Proof of Lemma* 2. By the assumption that the algorithm decreases the objective function, we have

$$\frac{1}{2}\left\|\boldsymbol{X} - \boldsymbol{U}_k\boldsymbol{V}_k^{\mathrm{T}}\right\|_F^2 + \frac{\lambda}{2}\|\boldsymbol{U}_k - \boldsymbol{V}_k\|_F^2 \leq \frac{1}{2}\left\|\boldsymbol{X} - \boldsymbol{U}_0\boldsymbol{U}_0^{\mathrm{T}}\right\|_F^2$$

which further implies that

$$\begin{cases} \left\|\boldsymbol{X} - \boldsymbol{U}_k\boldsymbol{V}_k^{\mathrm{T}}\right\|_F \leq \left\|\boldsymbol{X} - \boldsymbol{U}_0\boldsymbol{U}_0^{\mathrm{T}}\right\|_F \\ \frac{\lambda}{2}\left(\|\boldsymbol{U}_k\|_F^2 + \|\boldsymbol{V}_k\|_F^2 - 2|\langle \boldsymbol{U}_k\boldsymbol{V}_k^T, \mathbf{I}_r\rangle|\right) \leq \frac{\lambda}{2}\|\boldsymbol{U}_k - \boldsymbol{V}_k\|_F^2 \leq \frac{1}{2}\left\|\boldsymbol{X} - \boldsymbol{U}_0\boldsymbol{U}_0^{\mathrm{T}}\right\|_F^2 \end{cases}$$

where the first line further gives that

$$\|\boldsymbol{U}_k\boldsymbol{V}_k^T\|_F \leq \|\boldsymbol{X} - \boldsymbol{U}_0\boldsymbol{V}_0^T\|_F + \|\boldsymbol{X}\|_F,$$

while the second line leads to

$$\begin{aligned} \|\boldsymbol{U}_k\|_F^2 + \|\boldsymbol{V}_k\|_F^2 \leq & \frac{1}{\lambda}\|\boldsymbol{X} - \boldsymbol{U}_0\boldsymbol{U}_0^T\|_F^2 + 2\|\boldsymbol{U}_k\boldsymbol{V}_k^T\|_F\|\mathbf{I}_r\|_F \\ = & \frac{1}{\lambda}\|\boldsymbol{X} - \boldsymbol{U}_0\boldsymbol{U}_0^T\|_F^2 + 2\sqrt{r}\|\boldsymbol{U}_k\boldsymbol{V}_k^T\|_F \\ \leq & \left(\frac{1}{\lambda} + 2\sqrt{r}\right)\|\boldsymbol{X} - \boldsymbol{U}_0\boldsymbol{U}_0^T\|_F^2 + 2\sqrt{r}\|\boldsymbol{X}\|_F =: B_0 \end{aligned}$$

$\square$

**Theorem 2** (**Restatement of Theorem 2 in the original paper**). *Choose* $\lambda > \frac{1}{2}(\|\boldsymbol{X}\|_2 + \|\boldsymbol{X} - \boldsymbol{U}_0\boldsymbol{U}_0^T\|_F - \sigma_n(\boldsymbol{X}))$ *for* (3). *For any local search algorithm solving* (3) *with initialization* $\boldsymbol{V}_0 = \boldsymbol{U}_0$, *if it sequentially decreases the objective value, is convergent and converges to a critical point* $(\boldsymbol{U}^\star, \boldsymbol{V}^\star)$ *of* (3), *then we have* $\boldsymbol{U}^\star = \boldsymbol{V}^\star$ *and that* $\boldsymbol{U}^\star$ *is also a critical point of* (2).

*Proof of Theorem* 2. This theorem is a direct consequence of Theorem 1 and Lemma 2. $\square$

# 2 Main Convergence Results

## 2.1 ANLS for symmetric NMF(SymANLS)

---
**Algorithm 1** SymANLS
---
**Initialization:** $k = 1$ and $\boldsymbol{U}_0 = \boldsymbol{V}_0$.
 1: **while** stop criterion not meet **do**
 2:     $\boldsymbol{U}_k = \arg\min_{\boldsymbol{V} \geq 0} \frac{1}{2}\|\boldsymbol{X} - \boldsymbol{U}\boldsymbol{V}_{k-1}^T\|_F^2 + \frac{\lambda}{2}\|\boldsymbol{U} - \boldsymbol{V}_{k-1}\|_F^2$;
 3:     $\boldsymbol{V}_k = \arg\min_{\boldsymbol{U} \geq 0} \frac{1}{2}\|\boldsymbol{X} - \boldsymbol{U}_k\boldsymbol{V}^T\|_F^2 + \frac{\lambda}{2}\|\boldsymbol{U}_k - \boldsymbol{V}\|_F^2$;
 4:     $k = k + 1$.
 5: **end while**
**Output:** factorization $(\boldsymbol{U}_k, \boldsymbol{V}_k)$.
---

We restate Lemma 2, Theorem 3 and Corollary 1 in the original paper as follows, and provide their proof in Section 3.2.

**Lemma 3 (Restatement of Lemma 2 in the original paper).** *Let $\{(\boldsymbol{U}_k, \boldsymbol{V}_k)\}$ be the iterates sequence generated by Algorithm 1. Then we have*

$$f(\boldsymbol{U}_k, \boldsymbol{V}_k) - f(\boldsymbol{U}_{k+1}, \boldsymbol{V}_{k+1}) \geq \frac{\lambda}{2}(\|\boldsymbol{U}_{k+1} - \boldsymbol{U}_k\|_F^2 + \|\boldsymbol{V}_{k+1} - \boldsymbol{V}_k\|_F^2). \tag{13}$$

We now give the following main convergence guarantee for Algorithm 1.

**Theorem 3 (Restatement of Theorem 3 in the original paper).** *Let $\{(\boldsymbol{U}_k, \boldsymbol{V}_k)\}$ be the sequence generated by Algorithm 1. Then*

$$\lim_{k \to \infty} (\boldsymbol{U}_k, \boldsymbol{V}_k) = (\boldsymbol{U}^\star, \boldsymbol{V}^\star),$$

*where $(\boldsymbol{U}^\star, \boldsymbol{V}^\star)$ is a critical point of (3). Furthermore the convergence rate is at least sublinear.*

**Corollary 1 (Restatement of Corollary 1 in the original paper).** *Suppose Algorithm 1 is initialized with $\boldsymbol{V}_0 = \boldsymbol{U}_0$. Choose*

$$\lambda > \frac{1}{2}\left(\|\boldsymbol{X}\|_2 + \left\|\boldsymbol{X} - \boldsymbol{U}_0\boldsymbol{U}_0^T\right\|_F - \sigma_n(\boldsymbol{X})\right).$$

*Let $\{(\boldsymbol{U}_k, \boldsymbol{V}_k)\}$ be the sequence generated by Algorithm 1. Then $\{(\boldsymbol{U}_k, \boldsymbol{V}_k)\}$ is convergent and converges to $(\boldsymbol{U}^\star, \boldsymbol{V}^\star)$ with $\boldsymbol{U}^\star = \boldsymbol{V}^\star$ and $\boldsymbol{U}^\star$ a critical point of (2). Furthermore, the convergence rate is at least sublinear.*

*Proof of Corollary 1.* This corollary is a direct consequence of Theorem 2, Lemma 3 and Theorem 3. □

## 2.2 HALS for symmetric NMF (SymHALS)

---
**Algorithm 2** SymHALS
---
**Initialization:** $\boldsymbol{U}_0, \boldsymbol{V}_0$, iteration $k = 1$.

1: **while** stop criterion not meet **do**
2:    **for** $i = 1 : r$ **do**
3:       $\boldsymbol{X}_i^k = \boldsymbol{X} - \sum_{j=1}^{i-1} \boldsymbol{u}_j^k(\boldsymbol{v}_j^k)^{\mathrm{T}} + \sum_{j=i+1}^{r} \boldsymbol{u}_j^{k-1}(\boldsymbol{v}_j^{k-1})^{\mathrm{T}}$;
4:       $\boldsymbol{u}_i^k = \arg\min_{\boldsymbol{u}_i \geq \boldsymbol{0}} \frac{1}{2}\|\boldsymbol{X}_i^k - \boldsymbol{u}_i(\boldsymbol{v}_i^{k-1})^{\mathrm{T}}\|_F^2 + \frac{\lambda}{2}\|\boldsymbol{u}_i - \boldsymbol{v}_i^{k-1}\|_F^2 = \max\left(\frac{(\boldsymbol{X}_i^k + \lambda\mathbf{I})\boldsymbol{v}_i^{k-1}}{\|\boldsymbol{v}_i^{k-1}\|_2^2 + \lambda}, 0\right)$;
5:       $\boldsymbol{v}_i^k = \arg\min_{\boldsymbol{v}_i \geq \boldsymbol{0}} \frac{1}{2}\|\boldsymbol{X}_i^k - \boldsymbol{u}_i^k\boldsymbol{v}_i^{\mathrm{T}}\|_F^2 + \frac{\lambda}{2}\|\boldsymbol{u}_i^k - \boldsymbol{v}_i\|_F^2 = \max\left(\frac{(\boldsymbol{X}_i^k + \lambda\mathbf{I})\boldsymbol{u}_i^k}{\|\boldsymbol{u}_i^k\|_2^2 + \lambda}, 0\right)$;
6:    **end for**
7:    $k = k + 1$.
8: **end while**

**Output:** factorization $(\boldsymbol{U}_k, \boldsymbol{V}_k)$.

---

Algorithm SymHALS has similar descend property and convergence guarantee to algorithm SymANLS, three theoretical results are displayed in the following and their proof can be found in Section 3.3.

**Lemma 4.** *Suppose the iterates sequence $\{(\boldsymbol{U}_k, \boldsymbol{V}_k)\}$ is generated by Algorithm 2, then we have*

$$f(\boldsymbol{U}_k, \boldsymbol{V}_k) - f(\boldsymbol{U}_{k+1}, \boldsymbol{V}_{k+1}) \geq \frac{\lambda}{2}(\|\boldsymbol{U}_{k+1} - \boldsymbol{U}_k\|_F^2 + \|\boldsymbol{V}_{k+1} - \boldsymbol{V}_k\|_F^2).$$

**Theorem 4 (Sequence convergence of Algorithm 2).** *For any $\lambda > 0$, let $\{(\boldsymbol{U}_k, \boldsymbol{V}_k)\}$ be the sequence generated by Algorithm 2. Then*

$$\lim_{k \to \infty} (\boldsymbol{U}_k, \boldsymbol{V}_k) = (\boldsymbol{U}^\star, \boldsymbol{V}^\star)$$

*where $(\boldsymbol{U}^\star, \boldsymbol{V}^\star)$ is a critical point of (3). Furthermore the convergence rate is at least sublinear.*

**Corollary 2 (Restatement of Corollary 2 in the original paper).** *Suppose it is initialized with $\boldsymbol{V}_0 = \boldsymbol{U}_0$. Choose*

$$\lambda > \frac{1}{2}\left(\|\boldsymbol{X}\|_2 + \left\|\boldsymbol{X} - \boldsymbol{U}_0\boldsymbol{U}_0^T\right\|_F - \sigma_n(\boldsymbol{X})\right).$$

Let $\{(\boldsymbol{U}_k, \boldsymbol{V}_k)\}$ be the sequence generated by Algorithm 2. Then $\{(\boldsymbol{U}_k, \boldsymbol{V}_k)\}$ is convergent and converges to $(\boldsymbol{U}^\star, \boldsymbol{V}^\star)$ with

$$\boldsymbol{U}^\star = \boldsymbol{V}^\star$$

and $\boldsymbol{U}^\star$ a critical point of (2). Furthermore, the convergence rate is at least sublinear.

*Proof of Corollary 2.* This corollary is a direct consequence of Theorem 2, Lemma 4 and Theorem 4. □

# 3    Proof of Convergence Results

Proving Theorem 3 and Theorem 4 has very similar structure, and since the former is slightly easier to prove, hence we will give the detailed proof for Theorem 3 and provide a sketch to Theorem 4.

**Notation:** Define the iterates $\boldsymbol{W}_k = (\boldsymbol{U}_k, \boldsymbol{V}_k)$, in this remaining proofs, we may constantly change from $(\boldsymbol{U}_k, \boldsymbol{V}_k)$ to $\boldsymbol{W}_k$ for notational convenience.

We first give out the high level proof sketch.

**Proof sketch**: We prove sequence convergence by following the framework developed in [1,2]. The main contents are to establish sufficient decrease, safeguard, and a uniform Kurdyka-Lojasiewicz inequality for our proposed algorithms, which are stated in the following:

- *sufficient decrease.* $\exists C' > 0, s.t.$

$$f(\boldsymbol{W}_k) - f(\boldsymbol{W}_{k+1}) \geq C' \|\boldsymbol{W}_{k+1} - \boldsymbol{W}_k\|_F^2.$$

- and *safeguard.* $\exists C'' > 0, s.t.$

$$\exists\ \boldsymbol{B}_{k+1} \in \partial f(\boldsymbol{W}_{k+1}), \quad s.t.\ \|\boldsymbol{B}_{k+1}\|_F \leq C''\|\boldsymbol{W}_{k+1} - \boldsymbol{W}_k\|_F, \quad \forall$$

- uniform *Kurdyka-Lojasiewicz inequality* in Definition 3.

then the sequence convergence of $\{\boldsymbol{W}_k\}$ can be obtained by taking similar arguments as in [1,2].

Before going to the main convergence proof, we firstly introduce some supporting materials.

## 3.1    Definitions and basic ingredients

**Definition 1.** *The indicator function $\delta_+(\boldsymbol{x})$ of nonnegative constraint is defined as*

$$\delta_+(\boldsymbol{x}) = \begin{cases} 0, & \boldsymbol{x} \geq 0 \\ \infty, & otherwise \end{cases}$$

Since problem (3) is nonsmooth and nonconvex, we need some generalized differential to characterize its optimality.

**Definition 2.** *[1,3] Let $h : \mathbb{R}^d \to (-\infty, \infty]$ be a proper and lower semi-continuous function*

(i) *the effective domain is defined as*

$$\operatorname{dom} h := \left\{ \boldsymbol{u} \in \mathbb{R}^d : h(\boldsymbol{u}) < \infty \right\}.$$

(ii) *The (Fréchet) subdifferential $\partial h$ of $h$ at $\boldsymbol{u}$ is defined by*

$$\partial h(\boldsymbol{u}) = \left\{ \boldsymbol{z} : \liminf_{\boldsymbol{v} \to \boldsymbol{u}, v \neq u} \frac{h(\boldsymbol{v}) - h(\boldsymbol{u}) - \langle \boldsymbol{z}, \boldsymbol{v} - \boldsymbol{u} \rangle}{\|\boldsymbol{u} - \boldsymbol{v}\|} \geq 0 \right\}$$

*for any $\boldsymbol{u} \in \operatorname{dom} h$ and $\partial h(\boldsymbol{u}) = \emptyset$ if $\boldsymbol{u} \notin \operatorname{dom} h$.*

**Remark.** First, when $h(\boldsymbol{u})$ is differentiable at $\boldsymbol{u}$, the (Fréchet) subdifferential reduces to the simple derivative or gradient in multiple dimension. In this paper, we main consider the NMF problem (2), where the nonsmooth indicator function of the nonnegative constraint $\delta_+(\cdot)$ is subdifferentialble everywhere in its effective domain. Therefore, the subdiferential of the objective in (3) is simply given by: gradient of its smooth part + the subdifferential of the nonsmooth part $\delta_+(\cdot)$, where the '+' represents Minkowsiki summation of sets. Finally, the (Fréchet) subdifferential is commonly used to measure the optimality. In particular, a necessary condition for optimality is $0 \in \partial h(\overline{\boldsymbol{u}})$, and such a point is called critical point of $h(\boldsymbol{u})$.

By Definition 2, the subdifferential of the objective function $f(\boldsymbol{U}, \boldsymbol{V})$ is given in the following lemma.

**Lemma 5.** *The subdifferential of $f$ in Equation* (3) *is given by*

$$
\begin{aligned}
\partial_{\boldsymbol{U}} f(\boldsymbol{U}, \boldsymbol{V}) &= \nabla_{\boldsymbol{U}} g(\boldsymbol{U}, \boldsymbol{V}) + \partial \delta_+(\boldsymbol{U}), \\
\partial_{\boldsymbol{V}} f(\boldsymbol{U}, \boldsymbol{V}) &= \nabla_{\boldsymbol{V}} g(\boldsymbol{U}, \boldsymbol{V}) + \partial \delta_+(\boldsymbol{V}),
\end{aligned}
\tag{14}
$$

*where*

$$
\begin{aligned}
\partial \delta_+(\boldsymbol{U}) &= \left\{ \boldsymbol{S} \in \mathbb{R}^{n \times r} : \delta_+(\boldsymbol{U}') - \delta_+(\boldsymbol{U}) \geq \langle \boldsymbol{S}, \boldsymbol{U}' - \boldsymbol{U} \rangle, \ \forall \boldsymbol{U}' \in \mathbb{R}^{n \times r} \right\} \\
&= \left\{ \boldsymbol{S} \in \mathbb{R}^{n \times r} : \boldsymbol{S} \leq \boldsymbol{0}, \ \boldsymbol{S} \odot \boldsymbol{U} = \boldsymbol{0} \right\}
\end{aligned}
\tag{15}
$$

The following property states the geometry of objective function(including its constraints) around its critical points, which plays a key role in our sequel analysis.

**Definition 3** (**Kurdyka-Łojasiewicz (KL) property**). *[2,4] We say a proper semi-continuous function $h(\boldsymbol{u})$ satisfies Kurdyka-Łojasiewicz (KL) property, if $\overline{\boldsymbol{u}}$ is a critical point of $h(\boldsymbol{u})$, then there exist $\delta > 0$, $\theta \in [0, 1)$, $C_1 > 0$, s.t.*

$$
|h(\boldsymbol{u}) - h(\overline{\boldsymbol{u}})|^\theta \leq C_1 \operatorname{dist}(0, \partial h(\boldsymbol{u})), \quad \forall \ \boldsymbol{u} \in B(\overline{\boldsymbol{u}}, \delta)
$$

**Remark.** We mention that the above KL property(also known as KL inequality) states the regularity of $h(\boldsymbol{u})$ around its critical point $\boldsymbol{u}$ and the KL inequality trivially holds at non-critical point. A very large set of functions satisfy the KL inequality, as stated in [1, Theorem 5.1], for a proper lower semi-continuous function, it has KL property once it is semi-algebraic. And the semi-algebraic property of sets and functions is sufficiently general, including but never limited to any polynomials, any norm, quasi norm, $\ell_0$ norm, smooth manifold, etc. For more discussions and examples, see [1, 5].

## 3.2  Proof of Theorem 3

**Lemma 6** (**Lipschitz continuous gradient**). *The function*

$$
g(\boldsymbol{U}, \boldsymbol{V}) = \frac{1}{2} \|\boldsymbol{X} - \boldsymbol{U}\boldsymbol{V}^{\mathrm{T}}\|_F^2 + \frac{\lambda}{2} \|\boldsymbol{U} - \boldsymbol{V}\|_F^2
$$

*has Lipschitz continuous gradient with the Lipschitz constant as $3B + 2\lambda + \|\boldsymbol{X}\|_F$ in any bounded $\ell_2$-norm ball $\{(\boldsymbol{U}, \boldsymbol{V}) : \|\boldsymbol{U}\|_F^2 + \|\boldsymbol{V}\|_F^2 \leq B\}$ for any $B > 0$.*

*Proof of Lemma 6.* Denote $\boldsymbol{W}_k := (\boldsymbol{U}_k, \boldsymbol{V}_k)$ and similar notations apply to $\boldsymbol{W}$ and $g(\boldsymbol{W})$. To obtain the Lipschitz constant, it is equivalently to bound the spectral norm of the quadrature form of the Hessian $[\nabla^2 g(\boldsymbol{W})](\boldsymbol{D}, \boldsymbol{D})$ for any $\boldsymbol{D} := (\boldsymbol{D}_U, \boldsymbol{D}_V)$:

$$
\begin{aligned}
[\nabla^2 g(\boldsymbol{W})](\boldsymbol{D}, \boldsymbol{D}) &= \|\boldsymbol{U}\boldsymbol{D}_V^T + \boldsymbol{D}_U \boldsymbol{V}^T\|_F^2 - 2\langle \boldsymbol{X} - \boldsymbol{U}\boldsymbol{V}^{\mathrm{T}}, \boldsymbol{D}_U \boldsymbol{D}_V^T \rangle + \lambda \|\boldsymbol{D}_V - \boldsymbol{D}_U\|_F^2 \\
&\leq 2\|\boldsymbol{U}\|_F^2 \|\boldsymbol{D}_V\|_F^2 + 2\|\boldsymbol{V}\|_F^2 \|\boldsymbol{D}_U\|_F^2 + 2(\|\boldsymbol{X}\|_F + \|\boldsymbol{U}\boldsymbol{V}^{\mathrm{T}}\|_F) \underbrace{\|\boldsymbol{D}_U \boldsymbol{D}_V^T\|_F}_{\leq \|\boldsymbol{D}\|_F^2/2} + 2 \underbrace{(\lambda\|\boldsymbol{D}_U\|_F^2 + \lambda\|\boldsymbol{D}_V\|_F^2)}_{=\lambda\|\boldsymbol{D}\|_F^2} \\
&\leq (3\|\boldsymbol{U}\|_F^2 + 3\|\boldsymbol{V}\|_F^2 + \|\boldsymbol{X}\|_F + 2\lambda)\|\boldsymbol{D}\|_F^2 \leq (3B + \|\boldsymbol{X}\|_F + 2\lambda)\|\boldsymbol{D}\|_F^2.
\end{aligned}
$$

$\square$

**Remark.** As each iterate $(\boldsymbol{U}_k, \boldsymbol{V}_k)$ lives in the $\ell_2$-norm ball with the radius as $\sqrt{B_0}$ by (12). We immediately have the function $g(\boldsymbol{W})$ has Lipschitz continuous gradient with the Lipschitz constant being $2B_0 + \lambda + \|\boldsymbol{X}\|_F$ around each $\boldsymbol{W}_k = (\boldsymbol{U}_k, \boldsymbol{V}_k)$.

We now show the descend property in the following lemma.

**Proof of Lemma 3** Since all $\{(\boldsymbol{U}_k, \boldsymbol{V}_k)\}$ generated by Algorithm 1 are nonnegative, we identify that

$$
\begin{aligned}
f(\boldsymbol{U}_k, \boldsymbol{V}_k) - f(\boldsymbol{U}_{k+1}, \boldsymbol{V}_{k+1}) &= g(\boldsymbol{U}_k, \boldsymbol{V}_k) - g(\boldsymbol{U}_{k+1}, \boldsymbol{V}_{k+1}) \\
&= g(\boldsymbol{U}_k, \boldsymbol{V}_k) - g(\boldsymbol{U}_k, \boldsymbol{V}_{k+1}) + g(\boldsymbol{U}_k, \boldsymbol{V}_{k+1}) - g(\boldsymbol{U}_{k+1}, \boldsymbol{V}_{k+1})
\end{aligned} \tag{16}
$$

with $g$ denoting the smooth differentiable part of $f$.

The main proof would consist of bounding $(g(\boldsymbol{U}_k, \boldsymbol{V}_k) - g(\boldsymbol{U}_k, \boldsymbol{V}_{k+1}))$ and $(g(\boldsymbol{U}_k, \boldsymbol{V}_{k+1}) - g(\boldsymbol{U}_{k+1}, \boldsymbol{V}_{k+1}))$, respectively. We now bound $(g(\boldsymbol{U}_k, \boldsymbol{V}_k) - g(\boldsymbol{U}_k, \boldsymbol{V}_{k+1}))$. For that end, we identify that

$$
\boldsymbol{V}_{k+1} = \arg \min_{\boldsymbol{V} \geq 0} g(\boldsymbol{U}_k, \boldsymbol{V})
$$

As the indicator function of nonnegative constraint $\sigma_+(\boldsymbol{V})$ convex subdifferentiable for all $\boldsymbol{V}$ in its effective domain including relative boundary, hence the following set valued subdifferential

$$
\partial \sigma_+(\widetilde{\boldsymbol{V}}) = \{\boldsymbol{S} \in \mathbb{R}^{n \times r} : \sigma_+(\boldsymbol{V}) \geq \sigma_+(\widetilde{\boldsymbol{V}}) + \langle \boldsymbol{S}, \boldsymbol{V} - \widetilde{\boldsymbol{V}} \rangle, \ \forall \boldsymbol{V} \in \mathbb{R}^{n \times r}\}
$$

is nonempty for all $\boldsymbol{V} \geq 0$.

Utilizing the nonnegativity of iterates $\boldsymbol{V} = \boldsymbol{V}_k$, $\widetilde{\boldsymbol{V}} = \boldsymbol{V}_{k+1}$ gives

$$
0 \geq \langle \boldsymbol{S}_{k+1}, \boldsymbol{V}_k - \boldsymbol{V}_{k+1} \rangle, \ \forall \boldsymbol{S}_{k+1} \in \partial \delta_+(\boldsymbol{V}_{k+1})
$$

Since the update means $\boldsymbol{V}_{k+1} = \arg \min_{\boldsymbol{V}} g(\boldsymbol{U}_k, \boldsymbol{V}) + \sigma_+(\boldsymbol{V})$, it can be seen from the first order optimality $\boldsymbol{0} \in \nabla_{\boldsymbol{V}} g(\boldsymbol{U}_k, \boldsymbol{V}_{k+1}) + \partial \sigma_+(\boldsymbol{V}_{k+1})$ that

$$
\nabla_{\boldsymbol{V}} g(\boldsymbol{U}_k, \boldsymbol{V}_{k+1}) + \boldsymbol{S}_{k+1} = \boldsymbol{0}
$$

multiplying $\boldsymbol{V}_k - \boldsymbol{V}_{k+1}$ on both sides in the above equation providing

$$
\langle \nabla_{\boldsymbol{V}} g(\boldsymbol{U}_k, \boldsymbol{V}_{k+1}), \boldsymbol{V}_k - \boldsymbol{V}_{k+1} \rangle + \langle \boldsymbol{S}_{k+1}, \boldsymbol{V}_k - \boldsymbol{V}_{k+1} \rangle = \boldsymbol{0}
$$

hence

$$
\langle \nabla_{\boldsymbol{V}} g(\boldsymbol{U}_k, \boldsymbol{V}_{k+1}), \boldsymbol{V}_k - \boldsymbol{V}_{k+1} \rangle \geq 0 \tag{17}
$$

Now combining Equation (17) and the Taylor expansion, we arrive at the desired result

$$
\begin{aligned}
g(\boldsymbol{U}_k, \boldsymbol{V}_k) &= g(\boldsymbol{U}_k, \boldsymbol{V}_{k+1}) + \langle \nabla_{\boldsymbol{V}} g(\boldsymbol{U}_k, \boldsymbol{V}_{k+1}), \boldsymbol{V}_k - \boldsymbol{V}_{k+1} \rangle \\
&\quad + \frac{1}{2} \int_0^1 \nabla_{\boldsymbol{V}\boldsymbol{V}}^2 g(\boldsymbol{U}_k, t\boldsymbol{V}_k + (1-t)\boldsymbol{V}_{k+1})[\boldsymbol{V}_k - \boldsymbol{V}_{k+1}, \boldsymbol{V}_k - \boldsymbol{V}_{k+1}] \, \mathrm{d}t \\
&\geq g(\boldsymbol{U}_k, \boldsymbol{V}_{k+1}) + \frac{\lambda}{2} \|\boldsymbol{V}_k - \boldsymbol{V}_{k+1}\|_F^2
\end{aligned}
$$

where the last inequality is from the fact that $g(\boldsymbol{U}, \boldsymbol{V})$ is strongly convex on variable $\boldsymbol{V}$ with modulus at least $\lambda$. This further implies

$$
g(\boldsymbol{U}_k, \boldsymbol{V}_k) - g(\boldsymbol{U}_k, \boldsymbol{V}_{k+1}) \geq \frac{\lambda}{2} \|\boldsymbol{V}_k - \boldsymbol{V}_{k+1}\|_F^2.
$$

Using similar argument, we have

$$
g(\boldsymbol{U}_k, \boldsymbol{V}_{k+1}) - g(\boldsymbol{U}_{k+1}, \boldsymbol{V}_{k+1}) \geq \frac{\lambda}{2} \|\boldsymbol{U}_k - \boldsymbol{U}_{k+1}\|_F^2.
$$

We end the proof by plugging the above two inequalities back to Equation (16).

**Lemma 7** (**Convergence of objective function values**). *Suppose the iterates sequence $\{(\boldsymbol{U}_k, \boldsymbol{V}_k)\}$ is generated by Algorithm 1, the following holds*

*(a) The objective function value sequence $\{f(\boldsymbol{U}_k, \boldsymbol{V}_k)\}$ is nonincreasing and it converges to some finite value:*

$$\lim_{k\to\infty} f(\boldsymbol{U}_k, \boldsymbol{V}_k) = f^\star \tag{18}$$

*for some nonnegative $f^\star$ depending on $(\boldsymbol{U}_0, \boldsymbol{V}_0)$.*

*(b) The difference between iterates sequence is convergent to zero, i.e.,*

$$\lim_{k\to\infty} \|\boldsymbol{U}_{k+1} - \boldsymbol{U}_k\|_F = 0, \qquad \lim_{k\to\infty} \|\boldsymbol{V}_{k+1} - \boldsymbol{V}_k\|_F = 0. \tag{19}$$

*Proof of Lemma 7.* Denote $\boldsymbol{W}_k := (\boldsymbol{U}_k, \boldsymbol{V}_k)$ and similar notations apply to $\boldsymbol{W}$ and $f(\boldsymbol{W})$. Then it follows from (13) that

$$\sum_{k=0}^{\infty} f(\boldsymbol{W}_k) - f(\boldsymbol{W}_{k+1}) \geq \frac{\lambda}{2} \sum_{k=0}^{\infty} \|\boldsymbol{W}_{k+1} - \boldsymbol{W}_k\|_F^2 \implies \sum_{k=0}^{\infty} \|\boldsymbol{W}_{k+1} - \boldsymbol{W}_k\|_F^2 \leq \frac{2f(\boldsymbol{W}_0)}{\lambda}$$

Now, we conclude the proof of (a) by identifying that the sequence $\{f(\boldsymbol{W}_k)\}$ is non-increasing and lower-bounded by zero. For proving (b), we recognize the series $\{\sum_{k=n}^{\infty} \|\boldsymbol{W}_{k+1} - \boldsymbol{W}_k\|_F^2\}_n$ is convergent, so $\lim_{k\to\infty} \|\boldsymbol{W}_{k+1} - \boldsymbol{W}_k\|_F = 0$. □

**Lemma 8** (**Bounded iterates**). *Suppose the iterates sequence $\{(\boldsymbol{U}_k, \boldsymbol{V}_k)\}$ is generated by Algorithm 1, then it lies in a bounded subset, i.e.,*

$$\|\boldsymbol{U}_k\|_F^2 + \|\boldsymbol{V}_k\|_F^2 \leq B_0$$

*with $B_0$ defined in Equation (12).*

*Proof of Lemma 8.* The proof directly follows from Equation (12) in Lemma 2 and the sufficient decrease property in Lemma 3. □

**Lemma 9** (**Bounded subdifferential**). *Suppose the iterates sequence $\{\boldsymbol{W}_k = (\boldsymbol{U}_k, \boldsymbol{V}_k)\}$ is generated by Algorithm 1, then there exist $\boldsymbol{S}_{k+1} \in \partial_{\boldsymbol{U}} f(\boldsymbol{U}_{k+1}, \boldsymbol{V}_{k+1})$ and $\boldsymbol{D}_{k+1} \in \partial_{\boldsymbol{V}} f(\boldsymbol{U}_{k+1}, \boldsymbol{V}_{k+1})$ such that*

$$\left\| \begin{bmatrix} \boldsymbol{S}_{k+1} \\ \boldsymbol{D}_{k+1} \end{bmatrix} \right\|_F \leq (2B_0 + \lambda + \|\boldsymbol{X}\|_F)\|\boldsymbol{W}_{k+1} - \boldsymbol{W}_k\|_F. \tag{20}$$

*Proof of Lemma 9.* First from the optimality of $\boldsymbol{V}_{k+1}$, we have

$$\boldsymbol{0} \in \nabla_{\boldsymbol{V}} g(\boldsymbol{U}_k, \boldsymbol{V}_{k+1}) + \partial \sigma_+(\boldsymbol{V}_{k+1})$$

and noting that

$$\partial_{\boldsymbol{V}} f(\boldsymbol{U}_{k+1}, \boldsymbol{V}_{k+1}) = \nabla_{\boldsymbol{V}} g(\boldsymbol{U}_{k+1}, \boldsymbol{V}_{k+1}) + \partial \sigma_+(\boldsymbol{V}_{k+1}),$$

we hence can choose $\boldsymbol{D}_{k+1}$ as follows

$$\boldsymbol{D}_{k+1} = \nabla_{\boldsymbol{V}} g(\boldsymbol{U}_{k+1}, \boldsymbol{V}_{k+1}) - \nabla_{\boldsymbol{V}} g(\boldsymbol{U}_k, \boldsymbol{V}_{k+1}).$$

Then by the Lipschitz property of $g$ in Lemma 6 and the boundedness property $\|\boldsymbol{U}_k\|_F^2 + \|\boldsymbol{V}_k\|_F^2 \leq B_0$ in Lemma 8, we have

$$\|\boldsymbol{D}_{k+1}\|_F \leq (2B_0 + \lambda + \|\boldsymbol{X}\|_F)\|\boldsymbol{U}_{k+1} - \boldsymbol{U}_k\|_F.$$

On the other hand, we actually can choose $\boldsymbol{S}_{k+1} = 0$ by recognizing that

$$\boldsymbol{U}_{k+1} = \arg\min_{\boldsymbol{U}} f(\boldsymbol{U}, \boldsymbol{V}_{k+1}).$$

Thus, we have $\|(\boldsymbol{S}_{k+1}, \boldsymbol{D}_{k+1})\|_F \leq (2B_0 + \lambda + \|\boldsymbol{X}\|_F)\|\boldsymbol{W}_{k+1} - \boldsymbol{W}_k\|_F.$

□

We denote $\mathcal{C}(\boldsymbol{W}_0)$ as the collection of the limit points of sequence $\{\boldsymbol{W}_k\}$ (which may depend on the initialization $\boldsymbol{W}_0$). The following two results demonstrate some useful properties and optimality of $\mathcal{C}(\boldsymbol{W}_0)$.

**Lemma 10.** *Suppose the iterates sequence $\{(\boldsymbol{U}_k, \boldsymbol{V}_k)\}$ is generated by Algorithm 1, then*

$$f(\boldsymbol{U}^\star, \boldsymbol{V}^\star) = f^\star, \quad \forall\ (\boldsymbol{U}^\star, \boldsymbol{V}^\star) \in \mathcal{C}(\boldsymbol{U}_0, \boldsymbol{V}_0).$$

*In other words, all the limiting points of $\mathcal{C}(\boldsymbol{U}_0, \boldsymbol{V}_0)$ share the same function value, that is equal to $f^\star$, the limit of the objective function value sequence of Algorithm 1.*

*Proof of Lemma 10.* We first extract an arbitrary convergent subsequence $\{\boldsymbol{W}_{k_m}\}_m$ with limit $\boldsymbol{W}^\star$. By the definition of the algorithm we have

$$\boldsymbol{U}_k \geq 0,\ \boldsymbol{V}_k \geq 0, \quad \forall\ k \geq 0.$$

Furthermore,

$$\boldsymbol{U}^\star \geq 0,\ \boldsymbol{V}^\star \geq 0.$$

Hence,

$$\lim_{m \to \infty} \delta_+(\boldsymbol{U}_{k_m}) = 0, \quad \lim_{m \to \infty} \delta_+(\boldsymbol{V}_{k_m}) = 0.$$

Taking limit on the sequence of the function values $\{f(\boldsymbol{W}_{k_m})\}$, we have

$$f^\star = \lim_{m \to \infty} f(\boldsymbol{W}_{k_m}) = \lim_{m \to \infty} g(\boldsymbol{W}_{k_m}) + \lim_{m \to \infty} (\delta_+(\boldsymbol{U}_{k_m}) + \delta_+(\boldsymbol{V}_{k_m})) = g(\lim_{m \to \infty} \boldsymbol{W}_{k_m}) = g(\boldsymbol{W}^\star) = f(\boldsymbol{U}^\star, \boldsymbol{V}^\star).$$

Here the first equality comes from Equation (18), the third equality is due to the continuity of the smooth part $g(\boldsymbol{W})$ in (3) and the last equality is because $\boldsymbol{U}^\star \geq 0, \boldsymbol{V}^\star \geq 0$. Therefore, the proof is concluded by noting that $\boldsymbol{W}^\star$ is an arbitrary limiting point in $\mathcal{C}(\boldsymbol{W}_0)$. □

**Lemma 11.** *Suppose the iterates sequence $\{\boldsymbol{W}_k\}$ is generated by Algorithm 1, then each element $\boldsymbol{W}^\star := (\boldsymbol{U}^\star, \boldsymbol{V}^\star) \in \mathcal{C}(\boldsymbol{W}^0)$ is a critical point of (3) and $\mathcal{C}(\boldsymbol{W}^0)$ is a nonempty, compact and connected set, and satisfies*

$$\lim_{k \to \infty} \mathrm{dist}(\boldsymbol{W}_k, \mathcal{C}(\boldsymbol{W}_0)) = 0.$$

*Proof of Lemma 11.* It follows from Lemma 7 and Lemma 9 that then there exist $\boldsymbol{S}_{k+1} \in \partial_{\boldsymbol{U}} f(\boldsymbol{U}_{k+1}, \boldsymbol{V}_{k+1})$ and $\boldsymbol{D}_{k+1} \in \partial_{\boldsymbol{V}} f(\boldsymbol{U}_{k+1}, \boldsymbol{V}_{k+1})$ such that $\|(\boldsymbol{S}_{k+1}, \boldsymbol{D}_{k+1})\|_F \leq (2B_0 + \lambda + \|\boldsymbol{X}\|_F)\|\boldsymbol{U}_{k+1} - \boldsymbol{U}_k\|_F$ with the right hand side converging to zero as $k$ goes to infinity, implying that

$$\lim_{k \to \infty} (\boldsymbol{S}_k, \boldsymbol{D}_k) = \boldsymbol{0}.$$

Since $\{\boldsymbol{W}_k\}$ is bounded by Lemma 8, by the Bolzano-Weierstrass theorem, we now take an arbitrary convergent subsequence $\{\boldsymbol{W}_{k_m}\}_m$ that converges to a point $\boldsymbol{W}^\star \in \mathcal{C}(\boldsymbol{W}^0)$. By Lemma 5, we have

$$\boldsymbol{S}_{k_m} = \nabla_{\boldsymbol{U}} g(\boldsymbol{U}_{k_m}, \boldsymbol{V}_{k_m}) + \overline{\boldsymbol{S}}_{k_m}, \quad \overline{\boldsymbol{S}}_{k_m} \in \partial \delta_+(\boldsymbol{U}_{k_m}).$$

Since $\lim_{m \to \infty} \boldsymbol{S}_{k_m} = \boldsymbol{0}, \lim_{m \to \infty} \boldsymbol{W}_{k_m} = \boldsymbol{W}^\star$, and $\nabla_{\boldsymbol{U}} g$ is continuous, we have $\{\overline{\boldsymbol{S}}_{k_m}\}$ is convergent. Denote by $\overline{\boldsymbol{S}}^\star = \lim_{m \to \infty} \overline{\boldsymbol{S}}_{k_m}$. By the definition of $\overline{\boldsymbol{S}}_{k_m} \in \partial \delta_+(\boldsymbol{U}_{k_m})$, for any $\boldsymbol{U}' \in \mathbb{R}^{n \times r}$, we have

$$\delta_+(\boldsymbol{U}') - \delta_+(\boldsymbol{U}_{k_m}) \geq \langle \overline{\boldsymbol{S}}_{k_m}, \boldsymbol{U}' - \boldsymbol{U}_{k_m} \rangle.$$

Since $\lim_{m \to \infty} \delta_+(\boldsymbol{U}_{k_m}) = \delta_+(\boldsymbol{U}^\star) = 0$, taking $m \to \infty$ for both sides of the above equation gives

$$\delta_+(\boldsymbol{U}') - \delta_+(\boldsymbol{U}^\star) \geq \langle \overline{\boldsymbol{S}}^\star, \boldsymbol{U}' - \boldsymbol{U}^\star \rangle.$$

Since the above equation holds for any $\boldsymbol{U}' \in \mathbb{R}^{n \times r}$, we have $\overline{\boldsymbol{S}}^\star \in \partial \delta_+(\boldsymbol{U}^\star)$ and thus $\boldsymbol{0} = \nabla_{\boldsymbol{U}} g(\boldsymbol{U}^\star, \boldsymbol{V}^\star) + \overline{\boldsymbol{S}}^\star \in \partial_{\boldsymbol{U}} f(\boldsymbol{W}^\star)$. With similar argument, we get $\boldsymbol{0} \in \partial_{\boldsymbol{V}} f(\boldsymbol{W}^\star)$ and thus

$$\boldsymbol{0} \in \partial f(\boldsymbol{W}^\star).$$

since $\boldsymbol{W}^\star$ is an arbitrary elements in $\mathcal{C}(\boldsymbol{W}_0)$, implies each limiting point $\boldsymbol{W}^\star = (\boldsymbol{U}^\star, \boldsymbol{V}^\star) \in \mathcal{C}(\boldsymbol{W}_0)$ is a critical point of (3).

Finally, by [1, Lemma 3.5] and identifying the sequence $\{\boldsymbol{W}_k\}$ is bounded and regular (i.e. $\lim_{k\to\infty} \|\boldsymbol{W}_{k+1} - \boldsymbol{W}_k\|_F = 0$), we obtain that the set of accumulation points $\mathcal{C}(\boldsymbol{W}_0)$ is a nonempty compact and connect set satisfying

$$\lim_{k\to\infty} \mathrm{dist}(\boldsymbol{W}_k, \mathcal{C}(\boldsymbol{W}_0)) = 0.$$

$\square$

**Lemma 12 (Uniform KL property of $f$).** *For arbitrary $(\boldsymbol{U}, \boldsymbol{V}) \in \mathcal{C}(\boldsymbol{U}_0, \boldsymbol{V}_0)$, we can uniformly find a set of constants $C_2 > 0, \delta > 0, \theta \in [0,1)$ such that*

$$[f(\boldsymbol{W}) - f(\boldsymbol{W}^\star)]^\theta \le C_2 \, \mathrm{dist}(0, \partial f(\boldsymbol{W}))$$

*for all $\boldsymbol{W}$ such that $\mathrm{dist}(\boldsymbol{W}, \mathcal{C}(\boldsymbol{W}_0)) \le \delta$.*

*Proof of Lemma 12.* It is easy and straightforward to identify that $f(\boldsymbol{U}, \boldsymbol{V})$ satisfies the KL inequality at every point in its effective domain. From Lemma 11 we have the set $\mathcal{C}(\boldsymbol{W}_0)$ is a compact and connected set. Hence we can find finitely many balls $B(\boldsymbol{W}_i, r_i)$ and their union to cover

$$\mathcal{D} = \{(\boldsymbol{W}) : \mathrm{dist}(\boldsymbol{W}, \mathcal{C}(\boldsymbol{W}_0)) \le \delta\}$$

where each $r_i$ is chosen such that the KL inequality holds true at each center and we can choose $c_i > 0, \theta_i \in [0,1)$ that

$$[f(\boldsymbol{W}) - f(\boldsymbol{W}_i)]^{\theta_i} \le c_i \, \mathrm{dist}(0, \partial f(\boldsymbol{W})), \quad \forall \, \boldsymbol{W} \in B(\boldsymbol{W}_i, r_i).$$

Hence it is straightforward to verify

$$[f(\boldsymbol{W}) - f(\boldsymbol{W}^\star)]^\theta \le C_2 \, \mathrm{dist}(0, \partial f(\boldsymbol{W})).$$

for all $(\boldsymbol{W})$ such that $\mathrm{dist}(\boldsymbol{W}, \mathcal{C}(\boldsymbol{W}_0)) \le \delta$, where $C_2 = \max\{c_i\}$ and $\theta = \max\{\theta_i\}$.

$\square$

We now turn to the remaining proof of Theorem 3.

**Global Convergence.** We first assume there exists a finite $\bar{k}$ for which $f(\boldsymbol{W}_{\bar{k}}) = f(\boldsymbol{W}^\star)$. Then by the convergence of the function values $\{f(\boldsymbol{W}_k)\}$ with limit $f(\boldsymbol{W}^\star)$ in Lemma 10, we have

$$f(\boldsymbol{W}^\star) = f(\boldsymbol{W}_{\bar{k}}) = f(\boldsymbol{W}_{\bar{k}+1}) = \cdots$$

Together with the sufficient decrease property in Lemma 3, this implies

$$\boldsymbol{W}_{\bar{k}} = \boldsymbol{W}_{\bar{k}+1} = \boldsymbol{W}_{\bar{k}+2} = \cdots$$

establishing the convergence of the iterates $\{\boldsymbol{W}_k\}$ in finite steps.

Thus in the following, we assume $f(\boldsymbol{W}_k) > f(\boldsymbol{W}^\star)$ for all $k > 0$. First of all, since $\lim_{k\to\infty} \mathrm{dist}(\boldsymbol{W}_k, \mathcal{C}(\boldsymbol{W}_0)) = 0$ by Lemma 11, there exists a $k_0$ for which $\mathrm{dist}(\boldsymbol{W}_k, \mathcal{C}(\boldsymbol{W}_0)) \le \delta$ for all $k \ge k_0$ and any fixed $\delta > 0$. Hence

$$[f(\boldsymbol{W}_k) - f(\boldsymbol{W}^\star)]^\theta \le C_2 \, \mathrm{dist}(0, \partial f(\boldsymbol{W}_k)), \quad \forall k \ge k_0. \tag{21}$$

In the subsequent analysis, we restrict to $k \ge k_0$. Construct a concave function $x^{1-\theta}$ for some $\theta \in [0,1)$ with domain $x > 0$. Obviously, by the concavity, we have

$$x_2^{1-\theta} - x_1^{1-\theta} \ge (1-\theta) x_2^{-\theta} (x_2 - x_1), \forall x_1 > 0, x_2 > 0$$

by replacing $x_1$ by $f(\boldsymbol{W}_{k+1}) - f(\boldsymbol{W}^\star)$ and $x_2$ by $f(\boldsymbol{W}_k) - f(\boldsymbol{W}^\star)$ and using the sufficient decrease property in Lemma 3, we have

$$
\begin{aligned}
[f(\boldsymbol{W}_k) &- f(\boldsymbol{W}^\star)]^{1-\theta} - [f(\boldsymbol{W}_{k+1}) - f(\boldsymbol{W}^\star)]^{1-\theta} \\
&\geq (1-\theta) \frac{f(\boldsymbol{W}_k) - f(\boldsymbol{W}_{k+1})}{[f(\boldsymbol{W}_k) - f(\boldsymbol{W}^\star)]^\theta} \\
&\geq \frac{\lambda(1-\theta)}{2C_2} \frac{\|\boldsymbol{W}_k - \boldsymbol{W}_{k+1}\|_F^2}{\operatorname{dist}(0, \partial f(\boldsymbol{W}_k))}, \\
&\geq \frac{\lambda(1-\theta)}{2C_2(2B_0 + \lambda + \|\boldsymbol{X}\|_F)} \frac{\|\boldsymbol{W}_k - \boldsymbol{W}_{k+1}\|_F^2}{\|\boldsymbol{W}_k - \boldsymbol{W}_{k-1}\|_F} \\
&= \frac{\lambda(1-\theta)}{2C_2(2B_0 + \lambda + \|\boldsymbol{X}\|_F)} \left( \frac{\|\boldsymbol{W}_k - \boldsymbol{W}_{k+1}\|_F^2}{\|\boldsymbol{W}_k - \boldsymbol{W}_{k-1}\|_F} + \|\boldsymbol{W}_k - \boldsymbol{W}_{k-1}\|_F - \|\boldsymbol{W}_k - \boldsymbol{W}_{k-1}\|_F \right) \\
&\geq \frac{\lambda(1-\theta)}{2C_2(2B_0 + \lambda + \|\boldsymbol{X}\|_F)} (2\|\boldsymbol{W}_k - \boldsymbol{W}_{k+1}\|_F - \|\boldsymbol{W}_k - \boldsymbol{W}_{k-1}\|_F)
\end{aligned}
$$

where the second inequality follows from Equation (21) and the third inequality is by Lemma 9. So,

$$
2\|\boldsymbol{W}_k - \boldsymbol{W}_{k+1}\|_F - \|\boldsymbol{W}_k - \boldsymbol{W}_{k-1}\|_F \leq \beta \left( [f(\boldsymbol{W}_k) - f(\boldsymbol{W}^\star)]^{1-\theta} - [f(\boldsymbol{W}_{k+1}) - f(\boldsymbol{W}^\star)]^{1-\theta} \right)
$$

with $\beta := \left( \frac{\lambda(1-\theta)}{2C_2(2B_0 + \lambda + \|\boldsymbol{X}\|_F)} \right)^{-1}$.

Summing the above inequalities up from some $\widetilde{k} > k_0$ to infinity yields

$$
\sum_{k=\widetilde{k}}^{\infty} \|\boldsymbol{W}_k - \boldsymbol{W}_{k+1}\|_F \leq \|\boldsymbol{W}_{\widetilde{k}} - \boldsymbol{W}_{\widetilde{k}-1}\|_F + \beta[f(\boldsymbol{W}_{\widetilde{k}}) - f(\boldsymbol{W}^\star)]^{1-\theta} \tag{22}
$$

implying

$$
\sum_{k=\widetilde{k}}^{\infty} \|\boldsymbol{W}_k - \boldsymbol{W}_{k+1}\|_F < \infty.
$$

Following some standard arguments one can see that

$$
\limsup_{t \to \infty, t_1, t_2 \geq t} \|\boldsymbol{W}_{t_1} - \boldsymbol{W}_{t_2}\|_F = 0
$$

which implies that the sequence $\{\boldsymbol{W}_k\}$ is Cauchy, and hence convergent. Hence, the limit point set $\mathcal{C}(\boldsymbol{W}_0)$ is singleton $\boldsymbol{W}^\star$, which is also a critical point of (3) by Lemma 11.

**Convergence Rate.** Towards that end, we first know from the above argument that $\{\boldsymbol{W}_k\}$ converges to some point $\boldsymbol{W}^\star$, i.e., $\lim_{k \to \infty} \boldsymbol{W}^k = \boldsymbol{W}^\star$. Then using Equation (22) and the triangle inequality, we obtain

$$
\|\boldsymbol{W}_{\widetilde{k}} - \boldsymbol{W}^\star\|_F \leq \sum_{k=\widetilde{k}}^{\infty} \|\boldsymbol{W}_k - \boldsymbol{W}_{k+1}\|_F \leq \|\boldsymbol{W}_{\widetilde{k}} - \boldsymbol{W}_{\widetilde{k}-1}\|_F + \beta[f(\boldsymbol{W}_{\widetilde{k}}) - f(\boldsymbol{W}^\star)]^{1-\theta} \tag{23}
$$

which indicates the convergence rate of $\boldsymbol{W}_{\widetilde{k}} \to \boldsymbol{W}^\star$ is at least as fast as the speed that $\|\boldsymbol{W}_{\widetilde{k}} - \boldsymbol{W}_{\widetilde{k}-1}\|_F + \beta[f(\boldsymbol{W}_{\widetilde{k}}) - f(\boldsymbol{W}^\star)]^{1-\theta}$ tends to 0. In particular, the second term $\beta[f(\boldsymbol{W}_{\widetilde{k}}) - f(\boldsymbol{W}^\star)]^{1-\theta}$ can be controlled by combining Lemma 12 and Lemma 9:

$$
\begin{aligned}
\beta[f(\boldsymbol{W}_{\widetilde{k}}) - f(\boldsymbol{W}^\star)]^{1-\theta} &\leq \beta[C_2 \operatorname{dist}(0, \partial f(\boldsymbol{W}_{\widetilde{k}}))]^{\frac{1-\theta}{\theta}} \\
&\leq \underbrace{\beta[C_2(2B_0 + \lambda + \|\boldsymbol{X}\|_F)]^{\frac{1-\theta}{\theta}}}_{:=\alpha} \|\boldsymbol{W}_{\widetilde{k}} - \boldsymbol{W}_{\widetilde{k}-1}\|_F^{\frac{1-\theta}{\theta}} \tag{24}
\end{aligned}
$$

Plugging (24) back to (23), we then have

$$\sum_{k=\widetilde{k}}^{\infty} \|\boldsymbol{W}_k - \boldsymbol{W}_{k+1}\|_F \le \|\boldsymbol{W}_{\widetilde{k}} - \boldsymbol{W}_{\widetilde{k}-1}\|_F + \alpha \|\boldsymbol{W}_{\widetilde{k}} - \boldsymbol{W}_{\widetilde{k}-1}\|_F^{\frac{1-\theta}{\theta}}.$$

We divide the following analysis into two cases based on the value of the KL exponent $\theta$.

- *Case I*: $\theta \in [0, \frac{1}{2}]$. This case means $\frac{1-\theta}{\theta} \ge 1$. We define $P_{\widetilde{k}} = \sum_{i=\widetilde{k}}^{\infty} \|\boldsymbol{W}_{i+1} - \boldsymbol{W}_i\|_F$,

$$P_{\widetilde{k}} \le P_{\widetilde{k}-1} - P_{\widetilde{k}} + \alpha \left[ P_{\widetilde{k}-1} - P_{\widetilde{k}} \right]^{\frac{1-\theta}{\theta}}. \tag{25}$$

Since $P_{\widetilde{k}-1} - P_{\widetilde{k}} \to 0$, there exists a positive integer $k_1$ such that $P_{\widetilde{k}-1} - P_{\widetilde{k}} < 1$, $\forall \widetilde{k} \ge k_1$. Thus,

$$P_{\widetilde{k}} \le (1 + \alpha)\, (P_{\widetilde{k}-1} - P_{\widetilde{k}}), \quad \forall \widetilde{k} \ge \max\{k_0, k_1\},$$

which implies that

$$P_{\widetilde{k}} \le \rho \cdot P_{\widetilde{k}-1}, \quad \forall \widetilde{k} \ge \max\{k_0, k_1\}, \tag{26}$$

where $\rho = \frac{1+\alpha}{2+\alpha} \in (0,1)$. This together with (23) gives the linear convergence rate

$$\|\boldsymbol{W}_k - \boldsymbol{W}^\star\|_F \le \mathcal{O}(\rho^{k-\overline{k}}), \ \forall \ k \ge \overline{k}. \tag{27}$$

where $\overline{k} = \max\{k_0, k_1\}$.

- *Case II*: $\theta \in (1/2, 1)$. This case means $\frac{1-\theta}{\theta} \le 1$. Based on the former results, we have

$$P_{\widetilde{k}} \le (1 + \alpha) \left[ P_{\widetilde{k}-1} - P_{\widetilde{k}} \right]^{\frac{1-\theta}{\theta}}, \quad \forall \widetilde{k} \ge \max\{k_0, k_1\}.$$

We now run into the same situation as in [2, Theorem 2](after equation (13)) and [6, Theorem 2](after equation (30)), hence following a similar argument gives

$$P_{\widetilde{k}}^{\frac{1-2\theta}{1-\theta}} - P_{\widetilde{k}-1}^{\frac{1-2\theta}{1-\theta}} \ge \zeta, \ \forall \ k \ge \overline{k}$$

for some $\zeta > 0$. Then repeating and summing up the above inequality from $\overline{k} = \max\{k_0, k_1\}$ to any $k > \overline{k}$, we can conclude

$$P_{\widetilde{k}} \le \left[ P_{\widetilde{k}}^{\frac{1-2\theta}{1-\theta}} + \zeta(\widetilde{k} - \overline{k}) \right]^{-\frac{1-\theta}{2\theta-1}} = \mathcal{O}\left( (\widetilde{k} - \overline{k})^{-\frac{1-\theta}{2\theta-1}} \right), \ \forall \ \widetilde{k} > \overline{k}.$$

Finally, the following sublinear convergence holds

$$\|\boldsymbol{W}_k - \boldsymbol{W}^\star\|_F \le \mathcal{O}\left( (k - \overline{k})^{-\frac{1-\theta}{2\theta-1}} \right), \ \forall \ k > \overline{k}. \tag{28}$$

We end this proof by commenting that both linear and sublinear convergence rate are closely related to the KL exponent $\theta$ at the critical point $\boldsymbol{W}^\star$.

## 3.3  Proof sketch of Lemma 4 and Theorem 4

The proof of Lemma 4 and Theorem 4 is highly similar to that of Lemma 3 and Theorem 3, respectively, hence we just outline the sketch here. The main difference is to update one column by one column in Algorithm 2, but the key feature that the objective function at each update is strongly convex doesn't change. Hence we can employ a similar argument as in Lemma 3 to obtain the sufficient decreasing property of HALS.

- *Case I*: update $\boldsymbol{u}_i$, i.e. $(\boldsymbol{u}_1^{k+1}, \cdots, \boldsymbol{u}_{i-1}^{k+1}, \boldsymbol{u}_i^k, \cdots, \boldsymbol{u}_r^k, \boldsymbol{V}_k) \to (\boldsymbol{u}_1^{k+1}, \cdots, \boldsymbol{u}_i^{k+1}, \boldsymbol{u}_{i+1}^k, \cdots, \boldsymbol{u}_r^k, \boldsymbol{V}_k)$, we have

$$f(\boldsymbol{u}_1^{k+1}, \cdots, \boldsymbol{u}_i^k, \cdots, \boldsymbol{u}_r^k, \boldsymbol{V}_k) - f(\boldsymbol{u}_1^{k+1}, \cdots, \boldsymbol{u}_i^{k+1}, \cdots, \boldsymbol{u}_r^k, \boldsymbol{V}_{k+1}) \geq \frac{\lambda}{2} \|\boldsymbol{u}_i^{k+1} - \boldsymbol{u}_i^k\|_2^2$$

- *Case II*: update $\boldsymbol{v}_i$, i.e. $(\boldsymbol{U}_{k+1}, \boldsymbol{v}_1^{k+1}, \cdots, \boldsymbol{v}_{i-1}^{k+1}, \boldsymbol{v}_i^k, \cdots, \boldsymbol{v}_r^k) \to (\boldsymbol{U}_{k+1}, \boldsymbol{v}_1^{k+1}, \cdots, \boldsymbol{v}_i^{k+1}, \boldsymbol{v}_{i+1}^k, \cdots, \boldsymbol{v}_r^k)$, we have

$$f(\boldsymbol{U}_{k+1}, \boldsymbol{v}_1^{k+1}, \cdots, \boldsymbol{v}_{i-1}^{k+1}, \boldsymbol{v}_i^k, \cdots, \boldsymbol{v}_r^k) - f(\boldsymbol{U}_{k+1}, \boldsymbol{v}_1^{k+1}, \cdots, \boldsymbol{v}_i^{k+1}, \boldsymbol{v}_{i+1}^k, \cdots, \boldsymbol{v}_r^k) \geq \frac{\lambda}{2} \|\boldsymbol{v}_i^{k+1} - \boldsymbol{v}_i^k\|_2^2$$

Unrolling the update from $\boldsymbol{u}_1$ to $\boldsymbol{v}_r$ and summing them up, we can get the same sufficient decreasing inequality shown in Lemma 3. And the remaining proofs can be done following a similar way as in Theorem 3.

# 4 Additional Experimental Results

In Figure 1 and Figure 2, we show similar results as Figure 3 in the paper for larger truncated dataset MNIST$_{train}$, MNIST$_{test}$. Similarly, in Figure 3, we run ADMM 5000 iterations on ORL dataset.

Figure 1: Image clustering quality on MNIST$_{train}$ dataset, here $n = 3147, r = 10$.

Figure 2: Image clustering quality on MNIST$_{test}$ dataset, here $n = 3147, r = 3$.

# 5 Efficient Implementation of SymHALS

Figure 3: Image clustering quality of ADMM on ORL dataset, here $n = 400, r = 40$. ADMM needs roughly 3500 iterations to reach its maximum clustering rate!

---

**Algorithm 3** SymHALS

---

**Initialization:** $\boldsymbol{U}_0, \boldsymbol{V}_0$, iteration $k = 1$.

1: precompute residual $\boldsymbol{X}_1^k = X - \boldsymbol{U}^{k-1}(\boldsymbol{V}^{k-1})^T$.
2: **while** stop criterion not meet **do**
3:     **for** $i = 1 : r$ **do**
4:         $\boldsymbol{X}_i^k = \boldsymbol{X}_i^k + \boldsymbol{u}_i^{k-1}(\boldsymbol{v}_i^{k-1})^T$
5:         $\boldsymbol{u}_i^k = \arg\min_{\boldsymbol{u}_i \geq \boldsymbol{0}} \frac{1}{2}\|\boldsymbol{X}_i^k - \boldsymbol{u}_i(\boldsymbol{v}_i^{k-1})^{\mathrm{T}}\|_F^2 + \frac{\lambda}{2}\|\boldsymbol{u}_i - \boldsymbol{v}_i^{k-1}\|_F^2 = \max\left(\frac{(\boldsymbol{X}_i^k + \lambda\mathbf{I})\boldsymbol{v}_i^{k-1}}{\|\boldsymbol{v}_i^{k-1}\|_2^2 + \lambda}, 0\right)$;
6:         $\boldsymbol{v}_i^k = \arg\min_{\boldsymbol{v}_i \geq \boldsymbol{0}} \frac{1}{2}\|\boldsymbol{X}_i^k - \boldsymbol{u}_i^k \boldsymbol{v}_i^{\mathrm{T}}\|_F^2 + \frac{\lambda}{2}\|\boldsymbol{u}_i^k - \boldsymbol{v}_i\|_F^2 = \max\left(\frac{(\boldsymbol{X}_i^k + \lambda\mathbf{I})\boldsymbol{u}_i^k}{\|\boldsymbol{u}_i^k\|_2^2 + \lambda}, 0\right)$;
7:         update residual as $\boldsymbol{X}_i^k = \boldsymbol{X}_i^k - \boldsymbol{u}_i^k(\boldsymbol{v}_i^k)^T$.
8:     **end for**
9:     $\boldsymbol{X}_i^{k+1} = \boldsymbol{X}_i^k$, $k = k + 1$.
10: **end while**

**Output:** factorization $(\boldsymbol{U}_k, \boldsymbol{V}_k)$.

---