[Reviews · NeurIPS 2018]

Reviewer 1



To cope with symmetric nonnegative matrix factorization with fast optimization algorithms, this paper transfer the traditional formulation into a simple framework with a splitting property, which makes efficient alternating-type algorithms feasible for solving the symmetric NMF problem. Based on ANLS and HALS for NMF, this paper proposes two optimization algorithms to solve the transfered problem in an alternating manner, and theoretically guarantees that critical points can be found by these algorithms with at least a sublinear convergence rate. This is an interesting and well-written paper, and the proposed algorithms are original and technically sound. Practical results on synthetic and real datasets demonstrate the efficiency of proposed algorithms in terms of convergence analysis and clustering accuracy, compared with popular algorithms for symmetric NMF. I have some minor concerns on this manuscript, and I would expect the author(s) to reconsider the following specific points before publication. 1. The norms used in the paper should be clearly defined. For example, what does the matrix L2-norm used in Theorem 2 mean? 2. As Corollary 1 and 2 shown, the algorithms converge to the critical point only if $\lambda$ is sufficiently large. Thus, it would be helpful for readers to better understand the behaviors of the algorithms by providing a sensitivity analysis of $\lambda$, rather than simply setting $\lambda=1$ in experiments. 3. Details of text and presentation can be improved. For example, Line 150. 'enable' -> 'enables'. Line 195. 'there do exist' -> 'there does exist'. Line 205. '+' -> '-'. There is a similar problem in Step 3 of Algorithm 2. Algorithm 2. In Steps 4 and 5, the squared L2-norm, rather than Frobenius norm, should be applied on the term $\mathbf{u}_i-\mathbf{v}_i$. Line 253. 'converge' -> 'converges'. Line 282. 'dataset' -> 'datasets'. Table 1. The result in the 2nd row of item TDT2 should be boldface.

Reviewer 2



Summary: The authors solve an important NMF problem namely the symmetric NMF which arises in a wide variety of real world problems such as clustering in domains such as images, document analysis. They propose to rewrite the objective as a regular NMF problem with an additional regularization term of requiring the two matrix factors to be the same. This enables them to now apply standard NMF alternating update algorithms such as ANLS and HALS to solve the symmetric NMF problem. Real-world results are shown by experiments on CBCL, ORL, MNIST datasets which obtain qualitatively good results and also are pretty fast. Comments: The problem is well-defined and the approach/results are clearly presented. Solving this efficiently is of great interest to the NMF community and also to ML in general because of its connections to other problems such as equivalence to K-means clustering. How does this differ from the PALM (2) approach? It seems like the key novelty is proving that the alternate updates converges to a critical point which is the same for (2) and (3). Proposed algorithms do not have any proximal terms unlike previous approaches but it would be good to highlight why that is important --- in theory/practice. For instance expanding on the point of exploiting the sufficient decreasing property could be helpful. Can we have other norms on the objective and or constraints on U and will the approach presented generalize to those settings? For instance like the sparse NMF problem considered in the PALM paper (say sparse symmetric NMF ). Also, it would be interesting to run the faster'' GCD algorithm (1). This could potentially show your approach to be much more efficient and make it more compelling. Does the proof have to depend on each base algorithm or are there properties they need to satisfy so that we don't have to redo the proof each time for each new algorithm? Lambda is chosen to be 1, can you show the "theory" value for the data sets? The fact that your algorithm is robust against a wide variety of datasets is interesting. Overall, I like the experimental results and theoretical contributions of the paper. I am a bit unclear with the novelty factor and hoping to clarify with further discussion. Papers: (1) Fast Coordinate Descent Methods with Variable Selection for Non-negative Matrix Factorization --- Cho-Jui Hsieh, Inderjit S. Dhillon KDD 2011 (2) Proximal Alternating Linearized Minimization for Nonconvex and Nonsmooth Problems --- Jerome Bolte, Shoham Sabach and Marc Teboulle

Reviewer 3



[Summary] In this paper, an algorithm for symmetric NMF is proposed. The key idea of the proposed method is to optimize a matrix by optimizing two different matrices with a penalty term for the equivalence of those two matrices. The main contribution is to prove the relationship between the original and the modified optimization problems and an equivalent condition about the penalty weight. Computational experiments show some advantages. [Related work] Actually, the idea of variable splitting for symmetric NMF is not original, and the same algorithm has been proposed in [10]. The proposed formulation Eq.(3) is completely equivalent to Eq.(15) in [10]. Furthermore, the proposed alternating nonnegative least squares (ANLS) algorithms is the same as in [10]. [Strengths] Several theorems were proven to clarify the relationship between the original and the modified optimization problems. An equivalent condition about the penalty weight is provided. Sublinear convergence properties of ANLS and HALS algorithms were proven. [Weakness] 1) Originality is limited because the main idea of variable splitting is not new and the algorithm is also not new. 2) Theoretical proofs of existing algorithm might be regarded as some incremental contributions. 3) Experiments are somewhat weak: 3-1) I was wondering why Authors conducted experiments with lambda=1. According to Corollary 1 and 2 lambda should be sufficiently large, however it is completely ignored for experimental setting. Otherwise the proposed algorithm has no difference from [10]. 3-2) In Figure 3, different evaluations are shown in different dataset. It might be regarded as subjectively selected demonstrations. 3-3) I think clustering accuracy is not very significant because there are many other sophisticated algorithms, and initializations are still very important for nice performance. It show just the proposed algorithm is OK for some applications. [Minor points] -- comma should be deleted at line num. 112: "... + \lambda(u-v), = 0 ...". -- "close-form" --> "closed-form" at line num.195-196. --- after feedback --- I understand that the contribution of this paper is a theoretical justification of the existing algorithms proposed in [10]. In that case, the experimental validation with respect to the sensitivity of "lambda" is more important rather than the clustering accuracy. So Fig. 1 in feedback file is nice to add paper if possible. I think dropping symmetry is helpful, however it is not new idea that is already being used. So, it will not change anything in practice to use it. Furthermore, in recent years, almost all application researchers are using some application specific extensions of NMF such as sparse NMF, deep NMF, semi-NMF, and graph-regularized NMF, rather than the standard NMF. Thus, this paper is theoretically interesting as some basic research, but weak from an application perspective. Finally, I changed my evaluation upper one as: "Marginally below the acceptance threshold. I tend to vote for rejecting this submission, but accepting it would not be that bad."